# DLP: Dynamic Layerwise Pruning in Large Language Models

**Yuli Chen** [1]   **Bo Cheng** [1]   **Jiale Han** [2]   **Yingying Zhang** [1]   **Yingting Li** [1]   **Shuhao Zhang** [1]

## Abstract

Pruning has recently been widely adopted to reduce the parameter scale and improve the inference efficiency of Large Language Models (LLMs). Mainstream pruning techniques often rely on uniform layerwise pruning strategies, which can lead to severe performance degradation at high sparsity levels. Recognizing the varying contributions of different layers in LLMs, recent studies have shifted their focus toward non-uniform layerwise pruning. However, these approaches often rely on pre-defined values, which can result in suboptimal performance. To overcome these limitations, we propose a novel method called Dynamic Layerwise Pruning (DLP). This approach adaptively determines the relative importance of each layer by integrating model weights with input activation information, assigning pruning rates accordingly. Experimental results show that DLP effectively preserves model performance at high sparsity levels across multiple LLMs. Specifically, at 70% sparsity, DLP reduces the perplexity of LLaMA2-7B by 7.79 and improves the average accuracy by 2.7% compared to state-of-the-art methods. Moreover, DLP is compatible with various existing LLM compression techniques and can be seamlessly integrated into Parameter-Efficient Fine-Tuning (PEFT). We release the code[1] to facilitate future research.

---

[1]State Key Laboratory of Networking and Switching Technology, Beijing University of Posts and Telecommunications, Beijing, China [2]Hong Kong University of Science and Technology, Hong Kong, China. Correspondence to: Bo Cheng <chengbo@bupt.edu.cn>, Jiale Han <jialehan@ust.hk>.

*Proceedings of the 42nd International Conference on Machine Learning*, Vancouver, Canada. PMLR 267, 2025. Copyright 2025 by the author(s).

[1]The code is available at: https://github.com/ironartisan/DLP.

## 1. Introduction

Pruning (Jaiswal et al., 2023; Ma et al., 2023; Sun et al., 2024; Muralidharan et al., 2024; Cai et al., 2024; Men et al., 2024) has garnered significant attention in both academia and industry due to its ability to substantially reduce the parameter count of Large Language Models (LLMs) (OpenAI, 2023; Touvron et al., 2023a;b; Dubey et al., 2024). The core concept of pruning is to optimize resource utilization by eliminating redundant or less important parameters. SparseGPT (Frantar & Alistarh, 2023) implements a layer-by-layer and row-by-row greedy pruning strategy, ensuring that local optimizations have minimal impact on global performance. Recent studies (Xiao et al., 2023; Lee et al., 2023; Lin et al., 2024) highlight the pivotal role of outliers in LLMs. Although outliers constitute a small fraction of the model, they exert a disproportionately large influence on predictive accuracy. Building on the emergence of outlier features in LLMs (Puccetti et al., 2022; Lee et al., 2023; Lin et al., 2024), Wanda (Sun et al., 2024) introduces a novel approach to evaluate weight importance by integrating the absolute weight values with the norm of the corresponding input activations.

Although previous works (Frantar & Alistarh, 2023; Zhang et al., 2024; Sun et al., 2024) have achieved satisfactory performance, they fail to account for the varying importance of different layers within the model, instead assigning a uniform sparsity rate to all layers. This limitation leads to a significant performance drop under high sparsity conditions. Inspired by the presence of outliers, Outlier Weighed Layerwise Sparsity (OWL) (Yin et al., 2024) introduces a novel pruning paradigm that leverages the criticality of layers with a higher proportion of outliers. Based on the principle that layers with a higher proportion of outliers are more critical, OWL assigns different sparsity rates to each layer of LLMs. In comparison to uniform layerwise pruning (Zhu & Gupta, 2018), OWL demonstrates superior performance in preserving model accuracy. However, OWL still has certain limitations in practical applications. Its reliance on predefined criteria for outlier selection not only limits its adaptability to the dynamic needs of the model but also hinders the achievement of optimal performance.

To address the above issue, we compute the unimportance of each layer from an inverse perspective, which is then

transformed into the relative importance between layers. Based on the principle that layers with higher importance should have lower sparsity, we allocate layerwise sparsity rates. Some previous works (He et al., 2019; Zhang et al., 2023) use the median to identify redundant elements in a model, assuming that central elements can be replaced by other elements from the same layer. We demonstrate the effectiveness of the median in LLMs through three empirical studies. Additionally, we inherently place more emphasis on outliers. Due to the median's insensitivity to outliers (Huber et al., 2001), it provides a more accurate reflection of the central tendency of a layer when the weights contain outliers.

In this paper, we propose a novel Dynamic Layerwise Pruning (DLP) method. DLP adaptively determines the importance of each layer by combining model weights with input activation information, offering greater flexibility in sparsity allocation. Our goal is to determine the layerwise importance of LLMs, which we first derive by identifying the layerwise unimportance and then applying an inversion operation to obtain relative importance. Specifically, we begin by calculating the unimportance of each Transformer block based on the median of model weights and input activation values in the same layer. We then evaluate the relative unimportance across layers, which leads to the determination of the model's relative importance. Finally, pruning rates are assigned to each layer according to the principle that layers with higher importance should have lower sparsity. The pipeline of DLP is illustrated in Figure 1.

We conduct comprehensive experimental evaluations across multiple mainstream LLMs with varying parameter sizes (ranging from 7B to 30B) and architectures (e.g., LLaMA (Touvron et al., 2023a), Vicuna (Chiang et al., 2023), Mistral (Jiang et al., 2023)). The experimental results show that our method consistently outperforms the state-of-the-art LLM pruning techniques, particularly at high sparsity levels. For instance, at 70% sparsity, DLP reduces the perplexity of LLaMA2-7B by 7.79 and improves the average accuracy by 2.7%. When evaluated on the DeepSparse (Kurtic et al., 2023) inference engine, DLP achieves 2.8x-3.7x end-to-end acceleration on CPU with 70% - 90% sparsity. Furthermore, we find that a brief period of fine-tuning can restore the LLM to a reasonable range after high sparsity pruning.

As a general method, our approach can be applied to unstructured pruning, as well as $N : M$ sparsity (Sun et al., 2021) and structured pruning (Ma et al., 2023), consistently outperforming layerwise methods. Our method is orthogonal to quantization (Dettmers et al., 2022; Xiao et al., 2023; Balanca et al., 2024), and it can also be extended to Singular Value Decomposition (SVD) and Parameter-Efficient Fine-Tuning (PEFT) (Liu et al., 2022a), achieving substantial performance improvements.

Overall, the contributions of our work are as follows:

- We propose a novel method for measuring layerwise importance that does not rely on empirical values or model type. This method comprehensively considers both intra-layer and inter-layer elements to automatically determine the relative importance of each layer.

- We propose an effective method for unstructured pruning. Extensive experimental results consistently show that, at high sparsity levels, DLP not only outperforms state-of-the-art pruning techniques for LLMs but also achieves significant end-to-end speedups on CPU.

- Our method can not only be integrated with LLM compression techniques but also extended to PEFT.

## 2. Related Work

### 2.1. LLM Pruning

LLM Pruning aims to improve computational efficiency by reducing redundant parameters while preserving model performance as much as possible. This technique is primarily categorized into structured pruning (Ma et al., 2023; An et al., 2024) and unstructured pruning (Frantar & Alistarh, 2023; Dong et al., 2024; Li et al., 2024a). Structured pruning removes specific dimensions of parameters, significantly simplifying the model architecture and accelerating the inference process. For example, LLM-Pruner (Ma et al., 2023) uses sensitivity analysis and task-specific requirements to automatically identify and prune substructures with minimal impact on model performance, achieving efficient grouped structural optimization. Unstructured pruning, in contrast, operates at a finer granularity by directly removing individual weights to enhance sparsity. Representative methods include Magnitude (Jaiswal et al., 2023), SparseGPT (Frantar & Alistarh, 2023), and Wanda (Sun et al., 2024). Magnitude evaluates the importance of each weight based on its absolute value. SparseGPT performs layerwise local pruning and reconstructs losses to prune models to at least 50% sparsity in a single step, and Wanda determines which weights to prune by considering both weights and activations. Our work primarily focuses on unstructured pruning.

### 2.2. Layerwise Importance for Pruning

Layerwise importance has emerged as a powerful technique for pruning LLMs, enabling significant reductions in model size and computational cost while maintaining or even improving performance. SparseGPT (Frantar & Alistarh, 2023) and Wanda (Sun et al., 2024) uniformly apply the same sparsity to every layer of the model. However, these methods do not account for the varying importance of layers within the

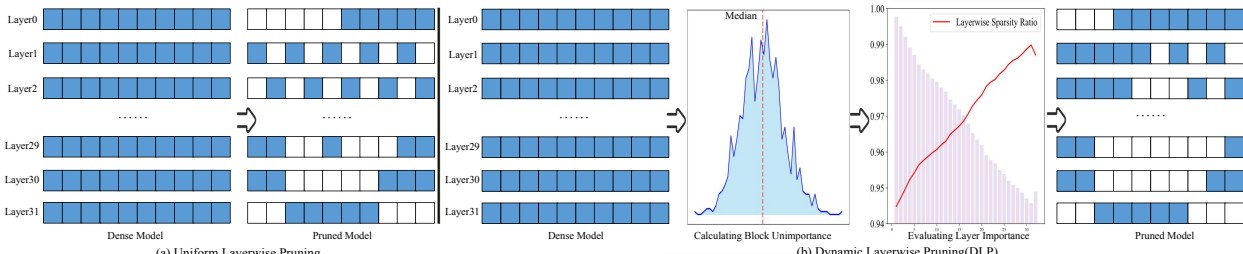

*Figure 1.* Illustration of Uniform Layerwise Pruning and Dynamic Layerwise Pruning (DLP): Blue squares represent unpruned weights, while white squares denote pruned weights. In uniform layerwise pruning, the same sparsity ratio is applied to every layer. In contrast, DLP calculates the unimportance of each Transformer block to compare the relative importance of layers, assigning different sparsity ratios based on the principle that layers with higher importance should have lower sparsity.

model, which can result in suboptimal performance. To overcome this limitation, recent studies (Gao et al., 2019; Yin et al., 2024) have explored non-uniform layerwise pruning approaches. Frankle & Carbin (2019) propose a uniform global threshold for pruning based on overall sparsity. Lee et al. (2021) introduce Layer-Adaptive Magnitude-based Pruning (LAMP), which dynamically determines the sparsity of each layer by calculating the relative importance of target connections. OWL (Yin et al., 2024) identifies a strong correlation between activation outliers in LLMs and their performance. By determining the sparsity ratio for each layer based on the proportion of outliers, OWL effectively preserves critical outliers. However, OWL defines outliers based on empirically set thresholds, which vary across models and may lead to suboptimal performance. In contrast, our method uses the median to automatically determine layerwise importance, adjusting the sparsity of each layer while maintaining the global sparsity rate.

### 2.3. Median in Pruning

The median is a statistical measure representing the middle value of a dataset when ordered in ascending or descending order. In recent years, median-based approaches have garnered significant attention in the field of model pruning. He et al. (2019) propose Filter Pruning via Geometric Median (FPGM), which leverages the geometric median to identify and prune redundant filters. Gkalelis & Mezaris (2020) utilize a geometric median-based criterion to identify and structurally prune the most redundant LSTM units. Zhang et al. (2023) introduce a pruning method based on Learned Representation Median (LRMF), which identifies unimportant filters by calculating the median in the frequency domain. Filters with values near the median are replaced with alternative representations, resulting in minimal impact on overall performance. However, the methods mentioned above all rely on the geometric median, which involves calculating the Euclidean distance to all points and

selecting the point with the smallest distance. This computation is complex and often approximated through iterative methods. In contrast, our work uses the median to measure the importance of layers, thereby reducing computational cost.

## 3. Methodology

In this section, we introduce our method, Dynamic Layerwise Pruning (DLP). We begin by defining the problem to be addressed, followed by a discussion of preliminary results. Next, we present three empirical studies to validate the proposed method, and finally, we provide a detailed description of the algorithm design.

### 3.1. Problem Definition

A popular solution strategy for model pruning is to decompose the task into multiple layerwise subproblems, enabling hierarchical optimization. These subproblems are often formulated as minimizing the $\ell_2$ error. Consider a neural network with $L$ layers, where the weight matrix $\mathbf{W}$ of shape $(C_{\text{out}}, C_{\text{in}})$, with $l \in 1, 2, \ldots, L$. $\mathbf{X}$ is the input activation with a shape of $(N \times L, C_{\text{in}})$, where $N$ and $L$ are batch size and sequence dimension, respectively. Specifically, for each layer $l$, the goal is to determine target weights $\hat{\mathbf{W}}_l$ that achieve a predefined pruning ratio $R$ while minimizing the squared error.

$$\text{argmin}_{\hat{\mathbf{w}}^l} \left\| \mathbf{W}^l \mathbf{X}^l - \hat{\mathbf{W}}^l \mathbf{X}^l \right\|_2^2 \qquad (1)$$

where $\mathbf{W}^l$ is the weight of the $l$-th layer, $\mathbf{X}^l$ is the input of the $l$-th layer, and $\|\cdot\|_2^2$ denotes $\ell_2$ norm squared.

### 3.2. Preliminaries

Inspired by the Optimal Brain Surgeon (Hassibi et al., 1993), SparseGPT (Frantar & Alistarh, 2023) quantifies the sen-

sitivity of the weights to the model error by means of the diagonal elements of the hessian matrix, with the less sensitive weights more suitable for pruning. The pruning metric of $l$-th layer in SparseGPT is:

$$\mathbf{E}_{ij}^l = \left[ |\mathbf{W}^l|^2 / \operatorname{diag}\left( \left( \mathbf{X}^{l\mathrm{T}}\mathbf{X}^l + \lambda \mathbf{I}^l \right)^{-1} \right) \right]_{ij} \quad (2)$$

where $|\cdot|^2$ represents the square of the absolute value operation, $\lambda$ is the damping term, which serves to prevent the values from becoming unstable, $\mathbf{I}^l$ is the unit matrix of the $l$-th layer, $\mathbf{X}^{l\mathrm{T}}\mathbf{X}^l + \lambda \mathbf{I}^l$ denotes the hessian matrix of the localized intra-layer reconstruction problem, $\operatorname{diag}\left( \left( \mathbf{X}^{l\mathrm{T}}\mathbf{X}^l + \lambda \mathbf{I}^l \right)^{-1} \right)$ denotes the diagonal element of the inverse of hessian matrix, indicating the reconstruction sensitivity of each connection, $i$ and $j$ represent the row and column indices of the matrix, respectively.

Wanda can be seen as a simplified version of SparseGPT, which avoids the complex computation of hessian matrix and reduces the pruning metric formula to a first-order approximation form, measuring the importance only by weights and input features. The score for the current weight of $l$-th layer is defined by:

$$\mathbf{A}_{ij}^l = \left| \mathbf{W}_{ij}^l \right| \cdot \left\| \mathbf{X}_j^l \right\|_2 \quad (3)$$

where $|\cdot|$ represents the absolute value operator, $\left\| \mathbf{X}_j^l \right\|_2$ evaluates the $\ell_2$ norm of the $j$-th feature vector in the input at layer $l$.

SparseGPT and Wanda both use the same sparsity rate for each layer and consider the contribution of each layer to be the same. However, these methods are not optimal for effectively allocating layerwise sparsity during the pruning of LLMs. Recently, OWL considers the retention rate of weight outliers and determines non-uniform sparsity rates across different layers based on the Layerwise Outlier Distribution (LOD). $\text{LOD} = \left[ D^1, D^2, \ldots, D^L \right]$, where $D^l$ characterizes the outlier distribution of $l$-th layer.

$$D^l = \frac{\sum_{i=1}^{C_{\text{out}}} \sum_{j=1}^{C_{\text{in}}} \mathbb{I}(\mathbf{A}_{ij}^l > \mathbf{M} \cdot \bar{\mathbf{A}}^l)}{C_{\text{in}} C_{\text{out}}} \quad (4)$$

where M is a constant, typically set to 5 or 7, $\bar{\mathbf{A}}^l$ is the mean of $\mathbf{A}^l$ and $\mathbb{I}(\cdot)$ is the indicator function, returning 1 if $\mathbf{A}_{ij}^l$ is larger than $\mathbf{M} \cdot \bar{\mathbf{A}}^l$, else 0.

However, M is determined empirically and is highly susceptible to variations in model parameters or types, which can result in suboptimal performance. In Figure 2, we investigate the performance relationship between model types, parameter scale, and the value of M. Notably, under the same

model type, the optimal M values differ between LLaMA1-7B and LLaMA1-13B. Similarly, for models with the same parameter scale, such as LLaMA1-7B and Vicuna-7B, the optimal M values also vary. This phenomenon indicates that the selection of M is influenced by both the model type and its parameter scale.

Moreover, using a fixed value may constrain the sparsity allocation to certain local patterns, overlooking the global importance distribution. To address this, we adopt a global perspective, assigning sparsity rates based on the relative importance of each layer. Pruning is carried out following the principle that the more important a layer is, the lower its sparsity rate. To validate our approach, we conduct three empirical studies based on the relative importance distribution.

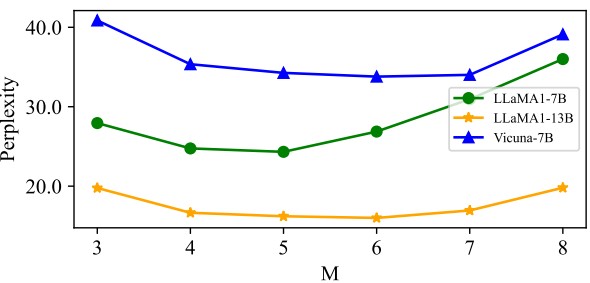

*Figure 2.* WikiText validation perplexity of LLaMA1-7B, LLaMA1-13B and Vicuna-7B pruned by various M at 70% sparsity using OWL.

### 3.3. Empirical Study

Relative Importance Distribution (RID). We use RID as the basis for assigning layer sparsity rates. RID takes into account both intra-layer and inter-layer element importance within LLMs. For intra-layer analysis, RID compares the absolute unimportance of Transformer blocks within the same layer. This is then converted into relative importance across layers, ultimately deriving the RID of the LLM. Specifically, for a given layer $l$, we use $\mathbf{A}_{ij}^l$ as the metric to evaluate the importance of weights. The unimportance score of $l$-th layer can be expressed as:

$$S^l = \sum_{i=1}^{C_{\text{out}}} \sum_{j=1}^{C_{\text{in}}} \mathbf{F}\left( \mathbf{A}_{ij}^l \right) \quad (5)$$

where $\mathbf{F}(\cdot)$ is a specific method, which is used to measure the absolute unimportance of the layer.

To make the importance across different layers comparable, we compute a normalized relative unimportance and convert

*Table 1.* WikiText validation perplexity of pruning metrics for LLaMA1-7B at 70% unstructured sparsity. The best performance result is indicated in bold.

| Method | Sum | Mean | Median | Max | Var | SD |
|---|---|---|---|---|---|---|
| Magnitude | 3.7e3 | 3.7e3 | **3.4e3** | 2.9e4 | 4.7e5 | 2.5e5 |
| SparseGPT | 18.24 | 18.24 | **17.76** | 38.57 | 21.33 | 21.42 |
| Wanda | 21.03 | 21.03 | **20.40** | 931.89 | 43.31 | 38.03 |

it into relative importance. For the $l$-th layer, the importance score is :

$$I^l = 1 - \frac{S^l}{\sum_{i=1}^l S^l} \qquad (6)$$

The importance scores of all layers constitute the RID, that is, $\text{RID} = \left[I^1, I^2, \ldots, I^L\right]$. Layers with lower importance have less impact on model performance, so they should be assigned higher sparsity.

**Empirical Study I: Evaluation of Unimportance Metrics.** FPGM (He et al., 2019) selects elements closest to the geometric median within a given layer for pruning, assuming that these elements are redundant and can be effectively represented by other elements in the same layer. To better evaluate the performance of pruning, we evaluate several common approaches, including Sum, Mean, Maximum (Max), Standard Deviation (SD) and Variance (Var). As presented in Table 1, the median method performs better than other methods. This shows its effectiveness in calculating layer unimportance. The median is more robust compared to other methods. It is less influenced by outliers. This allows it to better capture the performance of most layers. As a result, it calculates relative layer importance more accurately. Therefore, we choose $\mathbf{F}(\cdot)$ as the median.

**Empirical Study II: Relationship between Dense LLMs and Ours.** To investigate whether the proposed method can achieve non-uniform layer sparsity for dense LLMs, we use RID to measure the differences between layers in the LLMs. If RID is highly balanced, it indicates that our method is not suitable for evaluating inter-layer importance. As shown in the bar chart in the background of Figure 3, the results show that not all layers contribute equally to the model's performance. Surprisingly, this finding aligns closely with recent studies (Li et al., 2024c; Sun et al., 2025; Gromov et al., 2024), which show that deeper layers do not function as effectively as expected. Since the median is insensitive to extreme values, it better captures the central tendency. Elements near the center are easily represented by their neighboring elements, making their removal less detrimental to performance. A lower median within a layer suggests minimal redundancy in its weights, whereas a higher median implies greater redundancy. Consequently, layers with

*Table 2.* Comparison of OWL and Ours on LOD and Perplexity with LLaMA1-13B on the WikiText dataset at 70% unstructured sparsity. The best performance result is indicated in bold.

| Method | Layerwise Sparsity | LOD(%)↑ | Perplexity↓ |
|---|---|---|---|
| Dense | - | 5.43 | 5.09 |
| Magnitude | Uniform | 60.03 | 84511.48 |
| | OWL | 64.70 | 18992.87 |
| | Ours | **77.58** | **7642.99** |
| SparseGPT | Uniform | 47.70 | 18.93 |
| | OWL | 51.97 | 14.02 |
| | Ours | **64.46** | **12.63** |
| Wanda | Uniform | 55.14 | 56.26 |
| | OWL | 56.30 | 16.23 |
| | Ours | **70.06** | **13.65** |

higher redundancy are considered less influential to the overall model and are assigned a higher sparsity rate during pruning. Moreover, the overall trend of increasing sparsity suggests that earlier layers are considered more important, likely because they play a fundamental role in capturing low-level and generalizable features. In contrast, deeper layers tend to focus on more specialized or task-specific information, which may be more redundant or more tolerant to pruning (Fan et al., 2024). This further highlights the necessity of non-uniform layerwise sparsity.

**Empirical Study III: Comparison between OWL and Ours.** OWL aligns the sparsity ratio with the outlier ratio in each layer to preserve outliers. It also defines LOD as the ratio of the number of outlier weights to the total number of weights, including both zero and non-zero weights. We prune the LLaMA1-13B model using uniform layerwise pruning, OWL, and Ours, and compare the LOD after pruning. Following OWL's setup, We set M to 7. As shown in Table 2, our method achieves the highest LOD and the lowest perplexity. These results indicate that the proposed method outperforms OWL. Compared to uniform layerwise pruning, OWL and our method increase the proportion of outliers. Our method preserves outliers effectively even at high sparsity rates, maintaining better performance.

### 3.4. Dynamic Layerwise Pruning (DLP)

Although we obtain the RID, its scale is still influenced by the sparsity level. Therefore, it is necessary to further explore the relationship between sparsity levels and the scale of importance. Following the principle that layers with higher importance should have lower sparsity, we introduce a hyperparameter $\alpha$ to adjust this relationship. To mitigate the risk of severe performance degradation due to excessive pruning in a specific layer, we compress the range of importance into $[0, 2\alpha]$. Consequently, the sparsity of each

layer varies between $[R - \alpha, R + \alpha]$, with an overall average sparsity of $R$. The pseudocode of DLP is provided in Algorithm 1.

---

**Algorithm 1** Pseudocode of DLP

---

1: **Input:** Weight $\mathbf{W}$ and input $\mathbf{X}$
2: **Input:** Deflation scale $\alpha$ and sparsity rates $p$
3: **Output:** The dynamic pruning sparsity $R$ of each layer
4: Obtain the score $\mathbf{A}_{ij}^l$ via Eq.(3)
5: Obtain the unimportance score $S^l$ via Eq.(5)
6: Obtain the importance score $I^l$ via Eq.(6)
7: Store the scaled importance scores in $d$
8: **for** $i \leftarrow 0$ **to** length($I$) - 1 **do**
9: $\quad d_j \leftarrow \frac{I_i - I_{min}}{I_{max} - I_{min}} \times 2 \times \alpha$
10: **end for**
11: Calculate the average values $m$ of $d$
12: **for** $j \leftarrow 0$ **to** length($d$) - 1 **do**
13: $\quad R_j \leftarrow p + m - d_j$
14: **end for**
15: Obtain the final layerwise sparsity rates $R$
16: **Return** $R$

---

We compare the layerwise sparsity rates of OWL and DLP on LLaMA1 (7B/13B/30B) models. As shown in Figure 3, OWL and DLP exhibit similar overall trends. Notably, on the LLaMA1-30B model, DLP shows more significant fluctuations in layer sparsity rates. This indicates greater differences in relative importance between layers, resulting in a more fine-grained sparsity distribution across the model.

In addition, we investigate the relationship between pruning granularity and model performance in Appendix C. When allocating pruning rates, we choose to allocate them per layer rather than per Transformer block, as the former approach provides better performance. Furthermore, we also compare the performance of per-output pruning and per-layer pruning in Appendix D. During the actual pruning process, we perform pruning based on per output rather than per layer.

## 4. Experiments

### 4.1. Experimental Setup

**Models and Datasets.** We evaluate the performance of DLP on various LLMs, including LLaMA1 (7B/13B/30B) (Touvron et al., 2023a), LLaMA2 (7B/13B) (Touvron et al., 2023b), and other more advanced LLMs such as LLaMA3-8B (Dubey et al., 2024), LLaMA3.1-8B (Ling et al., 2024), Vicuna-7B (Chiang et al., 2023), Mistral-7B (Jiang et al., 2023), and Qwen-7B (Bai et al., 2023). These models are available from the HuggingFace Transformers library[2]. Additionally, we evaluate the language modeling capabilities

---
[2]https://github.com/huggingface/transformers

and zero-shot performance of sparse LLMs. Specifically, we measure language modeling performance using the perplexity metric on the WikiText (Merity et al., 2017), PTB (Marcus et al., 1994), and C4 (Raffel et al., 2020) validation datasets. For zero-shot evaluation, we assess accuracy on seven commonsense benchmarks from EleutherAI LM Harness (Gao et al., 2024), including BoolQ (Clark et al., 2019), RTE (Wang et al., 2019), HellaSwag (Zellers et al., 2019), WinoGrande (Sakaguchi et al., 2020), ARC Easy and Challenge (Boratko et al., 2018), and OpenBookQA (Mihaylov et al., 2018).

**Baselines.** We apply DLP to three LLM pruning methods: Magnitude (Jaiswal et al., 2023), SparseGPT (Frantar & Alistarh, 2023), and Wanda (Sun et al., 2024). Magnitude is a simple yet effective baseline that retains weights with larger absolute values to maintain model performance. SparseGPT and Wanda are two strong LLM pruning baselines capable of preserving reasonable performance under 50% sparsity. Our focus is primarily on high sparsity levels, no less than 50%. All three baselines adopt uniform layerwise sparsity rates. Additionally, we compare DLP with OWL to validate the performance of the proposed method. Furthermore, we evaluate it against other layerwise pruning methods, including Global (Frankle & Carbin, 2019), ER (Mocanu et al., 2018), ER-Plus (Liu et al., 2022b), and LAMP (Lee et al., 2021).

**Implementation Details.** In our experimental setup, we utilize four NVIDIA A40 GPUs, each with 48 GB of memory. To ensure a fair comparison, we follow OWL (Yin et al., 2024) by randomly selecting 128 samples from the C4 dataset, with each sample containing 2048 tokens as calibration data. In Appendix G, we present the hyperparameter configurations for various sparsity levels.

### 4.2. Main results

**Language Modeling.** We report the performance of various LLM pruning methods on language modeling with WikiText dataset, as presented in Table 3. Additionally, we provide performance results for different sparsity levels, with details available in Appendix B. The results validate the effectiveness of DLP's layerwise sparsity strategy. When used in combination with other unstructured pruning methods, DLP (ours) consistently achieves the lowest perplexity values across all model sizes, outperforming Uniform and OWL. For example, when the sparsity rate is 70%, DLP combined with Magnitude pruning on the LLaMA2-13B model has a Perplexity of 52.41, which is significantly better than the baseline based on uniform layerwise pruning of 214.19. In addition, when combined with the SparseGPT and Wanda methods, DLP still achieves a lower Perplexity than OWL. When the sparsity is 70%, SparseGPT reduces

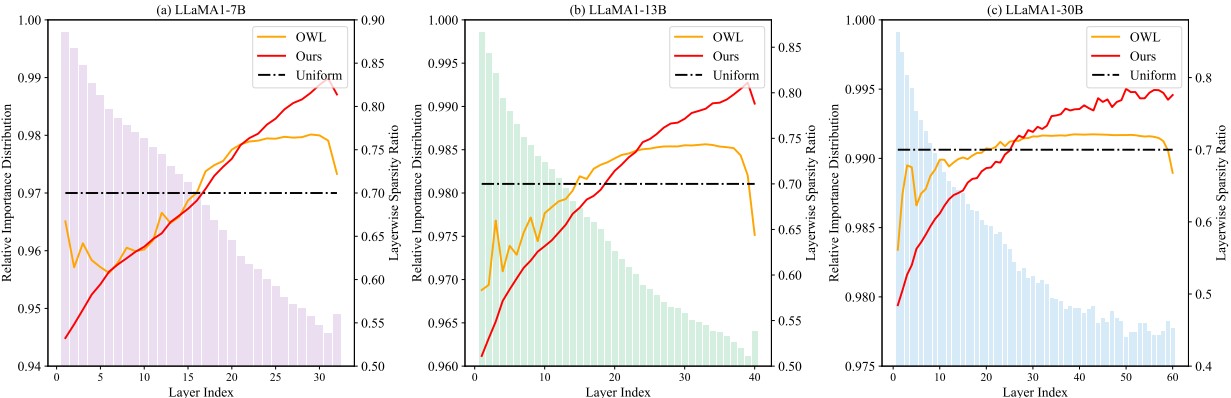

*Figure 3.* Comparison of layerwise sparsity distributions between Ours (red) and OWL (orange). The bar chart in the background represents the Relative Importance Distribution (RID). In each subplot, the horizontal axis represents the layer index, the left vertical axis corresponds to the RID, and the right vertical axis corresponds to the layerwise sparsity ratio.

*Table 3.* Perplexity results on WikiText. We produce the Uniform, OWL and DLP(Ours) with 70% unstructured sparsity on LLaMA1, LLaMA2 models. The best performance result is indicated in bold.

| Method | Layerwise Sparsity | LLaMA1 | | | LLaMA2 | |
|---|---|---|---|---|---|---|
| | | 7B | 13B | 30B | 7B | 13B |
| Dense | - | 5.68 | 5.09 | 4.10 | 5.47 | 4.88 |
| Magnitude | Uniform | 4.9e4 | 8.5e4 | 9.7e2 | 5.0e4 | 2.1e2 |
| | OWL | 2.0e4 | 1.9e4 | 2.4e2 | 1.5e4 | 57.55 |
| | Ours | **3.4e3** | **7.6e3** | **98.05** | **8.7e3** | **52.41** |
| SparseGPT | Uniform | 25.38 | 18.93 | 12.87 | 27.84 | 19.38 |
| | OWL | 19.95 | 14.02 | 10.22 | 19.71 | 15.12 |
| | Ours | **17.76** | **12.63** | **9.43** | **18.58** | **13.30** |
| Wanda | Uniform | 86.38 | 56.26 | 17.54 | 76.84 | 45.76 |
| | OWL | 24.46 | 16.23 | 10.77 | 30.58 | 20.65 |
| | Ours | **20.46** | **13.65** | **9.93** | **22.79** | **16.19** |

*Table 4.* Comparison of mean zero-shot accuracies (%) for pruned LLaMA1 and LLaMA2 models at 70% unstructured sparsity. The best performance result is indicated in bold.

| Method | Layerwise Sparsity | LLaMA1 | | | LLaMA2 | |
|---|---|---|---|---|---|---|
| | | 7B | 13B | 30B | 7B | 13B |
| Dense | - | 64.33 | 66.78 | 69.72 | 64.42 | 67.04 |
| Magnitude | Uniform | 34.80 | 37.09 | 35.06 | 35.65 | 36.23 |
| | OWL | 36.40 | 39.45 | 35.73 | 36.44 | 40.67 |
| | Ours | **38.21** | **40.11** | **42.03** | **40.84** | **44.02** |
| SparseGPT | Uniform | 45.32 | 48.34 | 55.87 | 44.72 | 47.99 |
| | OWL | 47.84 | 50.78 | 56.82 | 48.02 | 51.70 |
| | Ours | **48.32** | **53.06** | **57.84** | **49.65** | **53.47** |
| Wanda | Uniform | 39.91 | 41.62 | 54.59 | 37.04 | 40.44 |
| | OWL | 46.32 | 49.59 | 55.93 | 43.55 | 48.11 |
| | Ours | **48.62** | **52.03** | **56.83** | **46.25** | **51.11** |

it by 2.19 on LLaMA1-7B, and Wanda reduces it by 7.79 on LLaMA2-7B.

**Zero-Shot Tasks.** In Table 4, we present the average zero-shot accuracy of the pruned LLaMA1 and LLaMA2 models across seven zero-shot tasks. The performance for each specific task is provided in Appendix I. Notably, DLP consistently improves accuracy across all settings. For example, for the LLaMA2-13B model, DLP achieves average accuracy improvements of 7.79, 5.48, and 10.67 over Magnitude, Wanda, and SparseGPT, respectively. Compared to OWL, DLP improves the average accuracy by 3.35, 1.77, and 3.00. These results clearly demonstrate the potential of DLP in tackling more challenging zero-shot downstream tasks.

**Inference Speedup.** To verify the acceleration effect of sparse LLM after pruning by our method, we apply DLP

to LLaMA2-7B-chat-hf (Touvron et al., 2023b) for pruning using Wanda, and test its end-to-end decoding latency using the DeepSparse (Kurtic et al., 2023) inference engine on Intel(R) Xeon(R) Gold 6248R CPU equipped with 24 cores, and the results are shown in Table 5. The model pruned by DLP achieves significant inference speedup compared to the dense model. It is worth noting that the speedup ratio increases with sparsity and is 3.5x when the sparsity is 80%.

**Pruning Efficiency.** To evaluate the computational complexity of our method, we compare the empirical pruning speed with baselines. Specifically, since non-uniform layer sparsity can be pre-computed, we ignore the forward propagation and non-uniform sparsity calculation processes, focusing primarily on comparing the cumulative time spent on calculating pruning metrics for each layer between non-uniform and uniform layer pruning methods. The results

*Table 5.* End-to-end decoding latency and throughput of LLaMA2-7B-chat-hf on DeepSparse inference engine using DLP.

| Sparsity | Dense | 10% | 20% | 30% | 40% | 50% | 60% | 70% | 80% | 90% |
|---|---|---|---|---|---|---|---|---|---|---|
| Latency (ms) | 353.42 | 352.40 | 338.37 | 323.49 | 273.43 | 200.17 | 164.31 | 124.76 | 100.30 | 96.86 |
| Throughput (tokens/sec) | 2.83 | 2.84 | 2.96 | 3.09 | 3.66 | 4.99 | 6.08 | 8.01 | 9.97 | 10.32 |
| Speedup | | 1.0x | 1.0x | 1.1x | 1.1x | 1.3x | 1.8x | 2.2x | 2.8x | 3.5x | 3.7x |

*Table 6.* Comparison of time overhead used for computing the pruning metric across layers of LLaMA1-7B (in seconds).

| Method | Layerwise Sparsity | LLaMA1 7B | 13B | 30B |
|---|---|---|---|---|
| Magnitude | Uniform | 1.66 | 6.82 | 11.04 |
| | Ours | 1.68 | 6.65 | 11.01 |
| SparseGPT | Uniform | 254.32 | 511.30 | 1052.48 |
| | Ours | 257.48 | 487.44 | 1051.55 |
| Wanda | Uniform | 0.95 | 5.30 | 8.64 |
| | Ours | 0.98 | 5.50 | 8.50 |

*Table 7.* WikiText validation perplexity of various LLMs pruned by Ours with LoRA fine-tuning.

| Model | Method | Sparsity | Perplexity |
|---|---|---|---|
| LLaMA1-7B | Without FT | 0.7 | 17.76 |
| LLaMA1-7B | With FT | 0.7 | 12.15 |
| LLaMA1-13B | Without FT | 0.7 | 12.63 |
| LLaMA1-13B | With FT | 0.7 | 10.05 |

are shown in Table 6. In the case of uniform layer pruning, Wanda exhibits the lowest overhead compared to SparseGPT and Magnitude. As the number of model parameters increases, the efficiency of our method improves, with the time spent being lower than that of uniform layer pruning. This may be because our method aligns better with the model distribution, enabling faster identification of pruning targets.

**Fine-Tuning Performance.** In Table 7, we present the performance results after fine-tuning the model pruned with DLP. In order to expedite the model recovery process and improve its efficiency under limited data, we employ Low-Rank Adaptation (LoRA) (Hu et al., 2022) to post-train the pruned model. During fine-tuning, the pruning mask remains fixed, and the pretraining autoregressive loss is utilized. We fine-tune the LLaMA1-7B and LLaMA1-13B models pruned using SparseGPT on the C4 training dataset. The results indicate that the performance of highly sparse pruned models can be significantly restored with brief fine-tuning. The perplexity of LLaMA1-7B decreased by 5.61, while that of LLaMA1-13B decreased by 2.58.

*Table 8.* WikiText validation perplexity of various LLMs pruned by Uniform and Ours using Wanda. The best performance result is indicated in bold.

| Model | Method | 60% | 70% | 80% |
|---|---|---|---|---|
| LLaMA3-8B | Uniform | 23.50 | 122.96 | 687.11 |
| | Ours | **19.21** | **96.31** | **676.16** |
| LLaMA3.1-8B | Uniform | 21.84 | 118.18 | 1031.36 |
| | Ours | **18.58** | **84.30** | **786.19** |
| Vicuna-7B | Uniform | 12.89 | 60.60 | 1613.15 |
| | Ours | **11.49** | **28.79** | **345.10** |
| Mistral-7B | Uniform | 11.28 | 60.62 | 331.04 |
| | Ours | **9.91** | **29.83** | **199.19** |
| Qwen-7B | Uniform | 14.76 | 91.99 | 23136.98 |
| | Ours | **14.38** | **54.13** | **1122.54** |

### 4.3. More Corroborating Results of DLP

**Comparison among Various Layerwise Sparsity Methods.** To evaluate the superiority of the DLP method, we also compare the performance of DLP with other methods in terms of assigning layerwise sparsity in Appendix A. When the sparsity exceeds 40%, DLP consistently outperforms other layerwise sparsity methods. Notably, at a sparsity rate of 80%, the perplexity of DLP decreases by 56% compared to OWL.

**Performance on More Advanced LLMs.** To evaluate the applicability of DLP, we also assess its performance on more advanced LLMs, with the results presented in Table 8. Notably, as a general method, DLP is applicable to LLMs with different architectures at higher sparsity rates and consistently outperforms uniform layerwise pruning methods. The experimental results further confirm the effectiveness of DLP.

**Integration with Other Compression Methods.** In the previous sections, we focus on the combination of RID with unstructured pruning methods. To demonstrate the generality of the proposed method, we also combine RID with structured pruning methods such as LLM-Pruner, $N : M sparsity$, and quantization. In Appendix E.1, we investigate the performance of non-uniform layerwise structured pruning by combining RID with LLM-Pruner. In Appendix E.2, we investigate the application of RID within a hybrid $N : 8$ and $N : 4$ sparsity configuration (Sun et al.,

2021). In Appendix E.3, we integrate our method with SVD to improve the effectiveness of low-rank compression. In Appendix E.4, we examine the performance of the pruned model after applying GPTQ (Frantar et al., 2022).

**Integration with PEFT.**   Recent studies (Pan et al., 2024; Li et al., 2024b) highlight a significant imbalance in the distribution of weight norms across different layers in LoRA during fine-tuning tasks. By leveraging importance sampling across LLM layers and selectively freezing most intermediate layers during optimization, this approach improves fine-tuning performance while maintaining memory usage comparable to that of LoRA. We apply the RID to PEFT. In Appendix F, we compare the accuracy on few-shot tasks of LLaMA2-7B, evaluating our method against other PEFT methods. Notably, our method demonstrates significant performance improvements compared to current state-of-the-art approaches. It also further demonstrates the generalizability of our method.

## 5. Conclusion

In this paper, we propose a dynamic layerwise pruning method, DLP, which does not rely on empirical values or model architecture and can adaptively compute the relative importance of each layer. Specifically, we compute the median of each Transformer block within a layer to determine the absolute unimportance of the layer, which is then converted into the relative importance between layers. Layers with lower importance are assigned higher sparsity. Extensive experimental results show that our method consistently maintains excellent performance under high sparsity, significantly outperforming existing state-of-the-art methods. Notably, our approach demonstrates strong potential, not only being compatible with other compression techniques but also integrating effectively with PEFT.

## Acknowledgements

We would like to thank the anonymous reviewers for their thoughtful comments and support on this work. This work was supported in part by the National Key Research and Development Program of China under Grant 2022YFF0902701; in part by the National Natural Science Foundation of China under Grants U21A20468, 62372058, U22A2026.

## Impact Statement

This paper focuses on pruning LLMs. By automatically determining layerwise importance and assigning non-uniform sparsity, we can significantly reduce the number of parameters in LLMs at high sparsity rates while preserving their performance. Therefore, this advancement aids in the deployment of LLMs on resource-constrained devices, accelerates the inference process, and promotes the sustainable development of LLMs.

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

## A. Comparison among Various Layerwise Sparsity Methods

In Table 9, we further compare the performance of DLP with other layerwise sparsity methods on LLaMA1-7B. The details of these methods are as follows:

- Global (Frankle & Carbin, 2019). A global threshold is set across all layers to automatically adjust the sparsity of specific layers while ensuring the overall sparsity requirement is met.

- Uniform (Zhu & Gupta, 2018). Each layer is assigned the same sparsity rate for pruning.

- ER (Mocanu et al., 2018). The sparsity rate for each layer is proportional to $1 - \frac{c^{l-1}+c^l}{c^{l-1} \times c^l}$, where $c^l$ refers to the number of neurons/channels in the layer.

- ER-Plus (Liu et al., 2022b). Building on ER, the final layer is set as a dense layer if it is not, while maintaining the overall parameter count unchanged.

- LAMP (Lee et al., 2021). Layer-adaptive sparsity is achieved by calculating the importance score of each weight relative to other weights in the same layer and globally pruning the connections with the lowest scores.

- OWL (Yin et al., 2024). Non-uniform inter-layer sparsity is achieved by matching the sparsity rate of each layer with the proportion of outliers within that layer.

- AlphaPruning (Lu et al., 2024). The importance of each layer in an LLM is determined by analyzing the shape of the empirical spectral density of its weight matrix.

The results indicate that all methods perform well when the sparsity rate is below 40%. However, when the sparsity rate is greater than or equal to 40%, the performance differences become more noticeable. Notably, DLP and OWL perform exceptionally well at higher sparsity rates. When the sparsity rate reaches or exceeds 70%, DLP consistently outperforms OWL. At a sparsity rate of 70%, DLP reduces the perplexity by 4.00 compared to OWL.

*Table 9.* WikiText validation perplexity of LLaMA1-7B with various layerwise sparsity using Wanda. The best performance result is indicated in bold.

| Method | Sparsity(Dense 5.68) | | | | | | | |
| --- | --- | --- | --- | --- | --- | --- | --- | --- |
| | 10% | 20% | 30% | 40% | 50% | 60% | 70% | 80% |
| Global (Frankle & Carbin, 2019) | 14.11 | 3134 | 10293 | 10762 | 14848 | 17765 | 5147 | 39918.56 |
| Uniform (Zhu & Gupta, 2018) | 5.69 | 5.81 | 5.99 | 6.39 | 7.26 | 10.70 | 85.77 | 3499.88 |
| ER (Mocanu et al., 2018) | 5.69 | 5.80 | 6.02 | 6.55 | 7.74 | 12.16 | 112.03 | 11151.18 |
| ER-Plus (Liu et al., 2022b) | 5.70 | 5.82 | 6.05 | 6.62 | 8.00 | 14.04 | 229.17 | 6013.91 |
| LAMP (Lee et al., 2021) | 5.69 | **5.78** | **5.98** | 6.39 | 7.57 | 12.86 | 185.52 | 15647.87 |
| OWL (Yin et al., 2024) | 5.70 | 5.80 | 6.01 | 6.39 | 7.22 | 9.35 | 24.46 | 1227.24 |
| AlphaPruning (Lu et al., 2024) | 5.69 | 5.81 | 6.00 | **6.37** | 7.18 | 9.47 | 24.00 | 698.56 |
| Ours | 5.70 | 5.81 | 5.99 | 6.38 | **7.17** | 9.35 | **20.46** | **534.42** |

## B. Performance under Varying Levels of Sparsity

To evaluate the applicability of our method, we compare the perplexity of our method on the WikiText dataset with various sparsity rates, as presented in Table 10 and Figure 4. When sparsity is low, the differences among layerwise pruning methods are relatively minor, with SparseGPT demonstrating the best performance and Magnitude showing the worst. As sparsity increases, the perplexity differences among these methods become more pronounced. In Figure 4, it is clear that DLP outperforms other methods. When the sparsity is 70%, DLP reduces perplexity by 65.92 compared to uniform layerwise pruning, using Wanda. As shown in Table 10, when the sparsity is 80%, DLP reduces perplexity by 110.32 compared to uniform layerwise pruning and by 19.02 compared to OWL, using SparseGPT, while maintaining performance within a reasonable range. These results clearly demonstrate the superiority of our method.

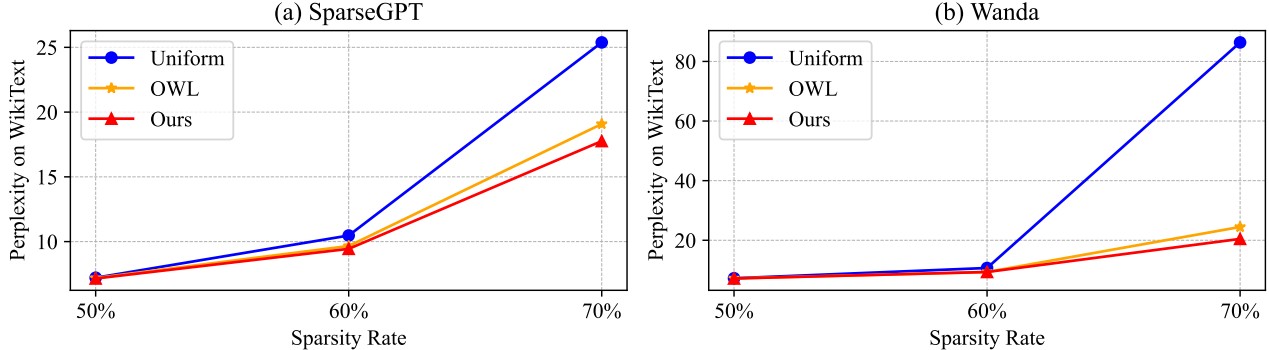

*Figure 4.* Comparison of different methods at high sparsity, using SparseGPT and Wanda.

*Table 10.* Perplexity results on WikiText. We produce the Uniform, OWL, and DLP with various unstructured sparsity rates on LLaMA1-7B model.

| Method | Layerwise Sparsity | Weight Update | Sparsity(Dense 5.68) | | | | | | | |
|---|---|---|---|---|---|---|---|---|---|---|
| | | | 10% | 20% | 30% | 40% | 50% | 60% | 70% | 80% |
| Magnitude | Uniform | $\times$ | 5.81 | 6.02 | 6.67 | 8.59 | 17.26 | 562.03 | 48834.17 | 132881.94 |
| | OWL | $\times$ | 8.11 | 6.01 | 6.55 | 8.13 | 13.86 | 82.75 | 19785.07 | 73458.74 |
| | Ours | $\times$ | 5.73 | 6.03 | 6.80 | 8.69 | 18.12 | 133.65 | 3437.55 | 48933.69 |
| SparseGPT | Uniform | $\sqrt{}$ | 5.70 | 5.80 | 5.96 | 6.32 | 7.21 | 10.47 | 25.38 | 180.94 |
| | OWL | $\sqrt{}$ | 5.71 | 5.81 | 6.00 | 6.32 | 7.19 | 9.66 | 19.08 | 89.64 |
| | Ours | $\sqrt{}$ | 5.70 | 5.81 | 5.97 | 6.32 | 7.15 | 9.44 | 17.76 | 70.62 |
| Wanda | Uniform | $\times$ | 5.70 | 5.82 | 6.00 | 6.39 | 7.26 | 10.70 | 86.38 | 7784.48 |
| | OWL | $\times$ | 5.70 | 5.80 | 6.01 | 6.39 | 7.22 | 9.35 | 24.46 | 1227.24 |
| | Ours | $\times$ | 5.70 | 5.81 | 5.99 | 6.38 | 7.17 | 9.35 | 20.46 | 534.42 |

## C. Per-Block vs. Per-Layer

As we mentioned before, we assign a distinct pruning ratio for each layer instead of each Transformer block. We test the performance of these methods on LLaMA1-7B under 70% sparsity, using DLP pruning with Wanda. The perplexity values are 3463.32 and 20.46, respectively. Table 11 and Table 12 present the sparsity levels of seven fully connected layers, including $q\_proj$, $k\_proj$, $v\_proj$, $o\_proj$, $gate\_proj$, $down\_proj$, and $up\_proj$ layers in layers 1, 2, 15, 30, and 31. It is noteworthy that comparing importance at the level of each Transformer block results in varying sparsity levels across blocks, leading to suboptimal performance. This is likely because such an approach creates significant sparsity discrepancies between blocks, potentially disrupting inter-layer information flow. In contrast, comparing importance at the layer level ensures uniform sparsity across Transformer blocks within each layer, which proves to be more beneficial for the performance of LLMs.

## D. Per-Output vs. Per-Layer

We also compare the performance of per-output pruning and per-layer pruning. As shown in Table 13 and Figure 5, we use Wanda to compare the perplexity of the LLaMA1-7B model at different sparsity levels for per-output pruning and per-layer pruning. Notably, the perplexity of the per-output pruning consistently outperforms per-layer pruning, with the performance gap becoming more pronounced as sparsity increases. Specifically, at 70% sparsity, the perplexity of the per-output method is 18.56 lower than that of the per-layer method. These results demonstrate that, after obtaining inter-layer importance through DLP, performing localized pruning on individual output neurons within each layer yields greater benefits.

*Table 11.* Sparsity of LLaMA1-7B pruned with per-layer DLP at 70% unstructured sparsity, using Wanda.

| Layer | q.proj | k.proj | v.proj | o.proj | gate.proj | down.proj | up.proj |
|---|---|---|---|---|---|---|---|
| 1 | 0.548 | 0.548 | 0.548 | 0.548 | 0.548 | 0.548 | 0.548 |
| 2 | 0.565 | 0.565 | 0.565 | 0.565 | 0.565 | 0.565 | 0.565 |
| 5 | 0.609 | 0.609 | 0.609 | 0.609 | 0.609 | 0.609 | 0.609 |
| 30 | 0.832 | 0.832 | 0.832 | 0.832 | 0.832 | 0.832 | 0.832 |
| 31 | 0.813 | 0.813 | 0.813 | 0.813 | 0.813 | 0.813 | 0.813 |

*Table 12.* Sparsity of LLaMA1-7B pruned with per-block DLP at 70% unstructured sparsity, using Wanda.

| Layer | q.proj | k.proj | v.proj | o.proj | gate.proj | down.proj | up.proj |
|---|---|---|---|---|---|---|---|
| 1 | 0.788 | 0.786 | 0.829 | 0.842 | 0.811 | 0.813 | 0.841 |
| 2 | 0.759 | 0.755 | 0.815 | 0.839 | 0.793 | 0.798 | 0.838 |
| 5 | 0.669 | 0.666 | 0.743 | 0.829 | 0.755 | 0.766 | 0.827 |
| 30 | 0.578 | 0.574 | 0.563 | 0.743 | 0.606 | 0.681 | 0.626 |
| 31 | 0.624 | 0.619 | 0.639 | 0.747 | 0.621 | 0.624 | 0.628 |

## E. Integration with Other Compression Methods

In the previous section, we primarily examine the effectiveness of combining our method with unstructured pruning methods. As a general non-uniform layerwise approach, our method is inherently applicable to a broader range of scenarios. To explore its potential, we apply RID to structured pruning, $N : M$ sparsity, SVD and quantization, respectively.

### E.1. Integration with Structured Pruning

In addition, we apply our method to structured pruning methods. Following the setup of LLM-Pruner (Ma et al., 2023), we prune not individual weights but entire neurons and attention heads. This approach directly reduces the model's parameter size and enables acceleration. We replace the uniform layerwise sparsity in LLM-Pruner with the non-uniform sparsity provided by DLP. The results, shown in Table 14, indicate that applying non-uniform sparsity allows LLM-Pruner to better preserve performance across different sparsity levels.

### E.2. Integration with $N : M$ Sparsity

To evaluate the potential of our method in hardware-friendly applications, we apply it to $N : M$ sparsity. Following the setup of DominoSearch (Sun et al., 2021), we investigate mixed $N : 8$ and $N : 4$ sparsity configurations. Unlike using a uniform $N$ value across all layers, we allocate different $N$ values based on layer importance while keeping the overall parameter count unchanged. The results, shown in Table 15, demonstrate that our method achieves superior performance compared to uniform $N : M$ sparsity. Notably, in high-sparsity scenarios of $1 : 4$ and $2 : 8$, our method reduces perplexity by 240x and 41x, respectively.

### E.3. Integration with SVD

We further extend our method to SVD to enhance low-rank compression. By leveraging RID, we assign different SVD compression rates to each layer. A higher RID score indicates greater layer importance, resulting in a lower compression rate to preserve model performance. In Table 16, we present the perplexity of LLaMA1-7B under various compression rates. As the compression rate increases, the model's performance degradation becomes more pronounced. Notably, our method consistently outperforms uniform layerwise SVD.

### E.4. Integration with Quantization

Finally, we apply the LLaMA1-7B model pruned with non-uniform layerwise sparsity to quantization techniques to evaluate whether it can maintain pre-pruning performance. Using the LLaMA1-7B model pruned to 70% sparsity with SparseGPT, we assess perplexity before and after quantization with GPTQ (Frantar et al., 2022) on the WikiText, PTB, and C4 datasets. The quantization bit widths are set to 3, 4, 8, and 16. The results, presented in Table 17 and Figure 6, reveal that the model

*Table 13.* Comparison of per-output and per-layer perplexity at different sparsity rates, using Wanda. The best performance result is indicated in bold.

| Method | 10% | 20% | 30% | 40% | 50% | 60% | 70% | 80% |
|---|---|---|---|---|---|---|---|---|
| Per-Output | 5.70 | **5.81** | **5.99** | **6.38** | **7.17** | **9.35** | **20.46** | **534.42** |
| Per-Layer | 5.70 | 5.82 | 6.03 | 6.56 | 7.70 | 10.95 | 39.02 | 886.10 |

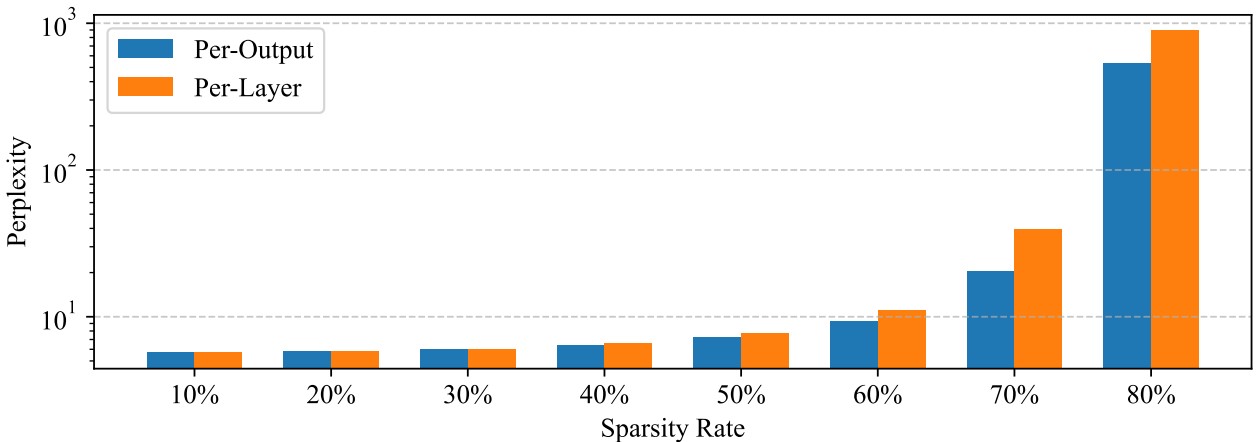

*Figure 5.* Comparison of per-output and per-layer perplexity at different sparsity rates. Due to the significant difference in values between high and low sparsity rates, we use the logarithmic results for comparison.

pruned with DLP consistently outperforms those with uniform sparsity during quantization. Notably, the performance of the 4-bit quantized model is nearly identical to that of the 16-bit quantized model. This demonstrates that applying DLP enables a 4x reduction in model size while maintaining performance.

## F. Integration with PEFT

We apply the RID to PEFT. We mainly consider two state-of-the-art baselines: Layerwise Importance Sampled AdamW (LISA) (Pan et al., 2024) and OwLore (Li et al., 2024b). LISA selectively updates critical LLM layers based on importance sampling while keeping the remaining layers unchanged. Inspired by OWL, OwLore strategically assigns higher sampling probabilities to layers with more outliers, selectively sampling only a few layers and fine-tuning their pre-trained weights. We assign higher sampling probabilities to layers with higher layerwise importance based on RID. Following the settings of OwLore, all methods are first fine-tuned on commonsense170k and then evaluated separately on different tasks with 5 shots, the results are shown in Table 18. Notably, our method achieves an average accuracy improvement of 4.88% compared to LISA and 0.8% compared to OwLore.

## G. Hyperparameter Setting

Due to the potentially wide range of importance differences across layers, directly using unadjusted importance scores may result in excessive pruning of some layers while under-pruning others. Adjusting the range of relative importance helps to make pruning decisions more balanced and precise across layers. To determine the matching relationship between different sparsity levels and relative importance, we conduct iterative experiments with $\alpha \in [0.02, 0.04, 0.06, 0.08, 0.1, 0.12, 0.15, 0.2]$. We provide the hyperparameter setting for different sparsity levels to facilitate the reproduction of our method's results, as shown in Table 19.

*Table 14.* Perplexity of structured pruning with LLaMA1-7B on WikiText. The best performance result is indicated in bold.

| Dataset | Method | Layerwise Sparsity | 20% | 40% | 60% | 80% |
|---------|--------|--------------------|------|------|------|------|
| WikiText | LLM-Pruner | Uniform | 18.61 | 647.20 | 4074.48 | 33849.77 |
| WikiText | LLM-Pruner | Ours | **16.62** | **29.62** | **115.58** | **642.16** |
| PTB | LLM-Pruner | Uniform | 90.02 | 538.65 | 2330.65 | 23447.05 |
| PTB | LLM-Pruner | Ours | **68.75** | **142.73** | **617.56** | **1932.18** |

*Table 15.* Perplexity of mixed N:M sparsity (N refers to non-zero weights) with LLaMA1-7B on WikiText. The best performance result is indicated in bold.

| Layerwise Sparsity | N:M Sparsity Structure | | | | | | | | | |
|--------------------|------|------|------|------|------|------|------|------|------|------|
| | 1:4 | 2:4 | 3:4 | 1:8 | 2:8 | 3:8 | 4:8 | 5:8 | 6:8 | 7:8 |
| Uniform | 7225.68 | 11.55 | 6.19 | 29790.41 | 3485.97 | 42.86 | 8.56 | 6.61 | 6.02 | 5.74 |
| Ours | **30.44** | **7.32** | **5.91** | **4990.69** | **83.50** | **10.42** | **7.19** | **6.27** | **5.87** | **5.71** |

# H. Robustness across Various Validation Datasets

To evaluate the robustness of the proposed method, we also test the performance of LLaMA1 and LLaMA2 models on different validation datasets. In Table 20, we report the perplexity of LLaMA1 (7B/13B/30B) pruned to 70% sparsity on WikiText, PTB, and C4. When using uniform layerwise sparsity, the Magnitude method performs the worst at high sparsity levels, while SparseGPT performs the best, primarily because SparseGPT updates weights post-pruning to recover performance. When combined with Wanda or SparseGPT, DLP consistently outperforms uniform layerwise pruning and OWL. In Table 21, we present the perplexity results of LLaMA2 (7B/13B) pruned to 70% sparsity on WikiText, PTB, and C4. Notably, DLP consistently outperforms the other layerwise methods. These experimental results strongly demonstrate the robustness of our method across different datasets.

# I. Zero-shot Tasks Performance

In Table 22 and Table 23, we present the accuracy of the pruned models on seven commonsense benchmarks from the EleutherAI LM Harness (Gao et al., 2024), including BoolQ (Clark et al., 2019), RTE(Wang et al., 2019), HellaSwag (Zellers et al., 2019), WinoGrande (Sakaguchi et al., 2020), ARC Easy and Challenge (Boratko et al., 2018), and OpenbookQA (Mihaylov et al., 2018). Notably, the average accuracy of our method consistently outperforms other layerwise methods.

*Table 16.* Perplexity of LLaMA1-7B across various compression rates. The best performance result is indicated in bold.

| Method | 0% | 10% | 20% | 30% | 40% | 50% |
|--------|-----|------|------|-------|---------|----------|
| Uniform | 5.68 | 6.58 | 8.41 | 17.69 | 1917.76 | 19170.02 |
| Ours | 5.68 | 6.58 | **8.38** | **15.33** | **1252.03** | **16304.26** |

*Table 17.* Perplexity of LLaMA1-7B on different validation datasets under varying quantization levels at 70% unstructured sparsity, pruned with DLP using SparseGPT. The best performance result is indicated in bold.

| Bits | Layerwise Sparsity | Sparsity | WikiText | PTB | C4 |
|------|--------------------|----------|----------|-----|-----|
| 16 | Dense | 0 | 5.68 | 31.50 | 7.08 |
| | Uniform | 70% | 26.49 | 298.15 | 24.57 |
| | Ours | 70% | **18.38** | **182.69** | **17.72** |
| 8 | Dense | 0 | 5.68 | 31.46 | 7.08 |
| | Uniform | 70% | 26.50 | 298.70 | 24.51 |
| | Ours | 70% | **18.38** | **183.25** | **17.71** |
| 4 | Dense | 0 | 5.79 | 32.16 | 7.23 |
| | Uniform | 70% | 27.08 | 315.35 | 24.93 |
| | Ours | 70% | **18.92** | **178.28** | **18.13** |
| 3 | Dense | 0 | 6.23 | 36.38 | 8.02 |
| | Uniform | 70% | 30.73 | 428.32 | 29.92 |
| | Ours | 70% | **21.56** | **243.98** | **20.59** |

*Table 18.* Accuracy(%) of different Parameter-Efficient Fine-Tuning (PEFT) methods on few-shot tasks with LLaMA2-7B. The best performance result is indicated in bold.

| Method | BoolQ | HellaSwag | WinoGrande | ARC-e | ARC-c | OBQA | Mean |
|--------|-------|-----------|------------|-------|-------|------|------|
| LISA (Pan et al., 2024) | 77.16 | 71.60 | 76.87 | 74.75 | 45.65 | 45.20 | 65.21 |
| OwLore (Li et al., 2024b) | **82.63** | 78.33 | 79.08 | 78.24 | 51.88 | 45.60 | 69.29 |
| Ours | 81.90 | **78.55** | **80.58** | **79.92** | **52.99** | **46.60** | **70.09** |

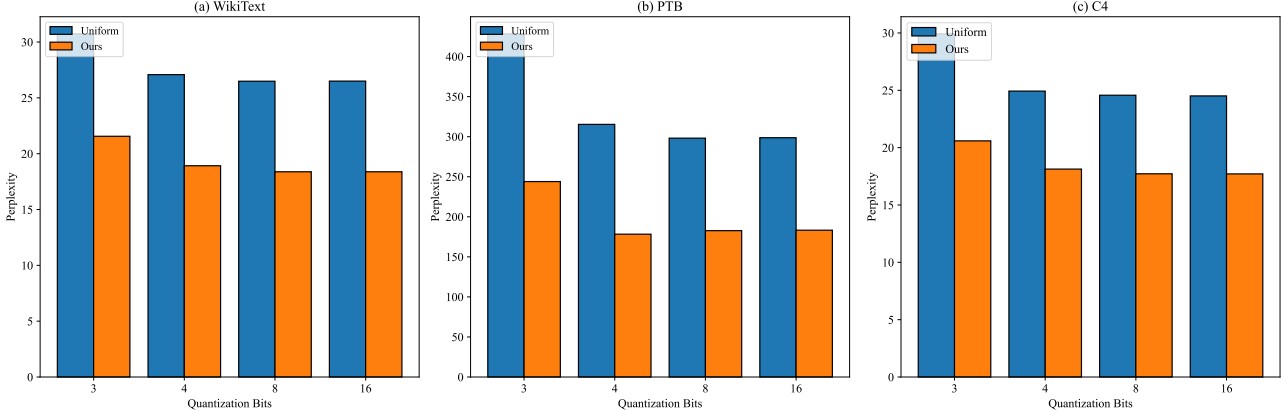

*Figure 6.* Perplexity of LLaMA1-7B on different validation datasets under varying quantization bits.

*Table 19.* Hyperparameter settings for different sparsity levels.

| Sparsity | 10% | 20% | 30% | 40% | 50% | 60% | 70% | 80% |
|----------|------|------|------|------|------|-----|------|------|
| $\alpha$ | 0.06 | 0.02 | 0.04 | 0.02 | 0.04 | 0.1 | 0.15 | 0.12 |

*Table 20.* Perplexity of LLaMA1 models on different validation datasets at 70% unstructured sparsity. The best performance result is indicated in bold.

| Model | Method | Layerwise Sparsity | Weight Update | WikiText | PTB | C4 |
|-------|--------|--------------------|---------------|----------|-----|-----|
| LLaMA1-7B | Dense | - | - | 5.68 | 41.15 | 7.34 |
| | Magnitude | Uniform | $\times$ | 48838.41 | 141244.92 | 23374.60 |
| | | OWL | $\times$ | 19785.07 | **164741** | 29854.54 |
| | | Ours | $\times$ | **3437.55** | 798602.88 | **9694.97** |
| | SparseGPT | Uniform | $\checkmark$ | 25.38 | 331.89 | 28.23 |
| | | OWL | $\checkmark$ | 19.08 | 221.95 | 20.61 |
| | | Ours | $\checkmark$ | **17.76** | **202.06** | **19.34** |
| | Wanda | Uniform | $\times$ | 86.38 | 698.63 | 88.14 |
| | | OWL | $\times$ | 24.46 | 398.58 | 27.36 |
| | | Ours | $\times$ | **20.46** | **302.15** | **22.71** |
| LLaMA1-13B | Dense | - | $\times$ | 5.09 | 28.10 | 6.80 |
| | Magnitude | Uniform | $\times$ | 84511.48 | 389981.53 | 40205.27 |
| | | OWL | $\times$ | 18992.87 | 61249.96 | **19282.38** |
| | | Ours | $\times$ | **7642.99** | **14510.26** | 33386.63 |
| | SparseGPT | Uniform | $\checkmark$ | 18.93 | 150.92 | 22.70 |
| | | OWL | $\checkmark$ | 14.02 | 99.02 | 16.19 |
| | | Ours | $\checkmark$ | **12.63** | **80.19** | **14.47** |
| | Wanda | Uniform | $\times$ | 56.26 | 324.01 | 54.69 |
| | | OWL | $\times$ | 16.23 | 146.35 | 18.84 |
| | | Ours | $\times$ | **13.65** | **100.94** | **15.82** |
| LLaMA1-30B | Dense | - | - | 4.10 | 23.51 | 6.13 |
| | Magnitude | Uniform | $\times$ | 971.71 | 10452.38 | 5372.17 |
| | | OWL | $\times$ | 242.80 | 2495.37 | 808.44 |
| | | Ours | $\times$ | **98.05** | **491.10** | **114.72** |
| | SparseGPT | Uniform | $\checkmark$ | 12.87 | 62.54 | 15.47 |
| | | OWL | $\checkmark$ | 10.22 | 48.42 | 12.79 |
| | | Ours | $\checkmark$ | **9.43** | **40.54** | **11.69** |
| | Wanda | Uniform | $\times$ | 17.54 | 111.33 | 18.81 |
| | | OWL | $\times$ | 10.77 | 60.24 | 13.62 |
| | | Ours | $\times$ | **9.93** | **46.51** | **12.36** |

*Table 21.* Perplexity of LLaMA2 models on different validation datasets at 70% unstructured sparsity. The best performance result is indicated in bold.

| Model | Method | Layerwise Sparsity | Weight Update | WikiText | PTB | C4 |
|---|---|---|---|---|---|---|
| | Dense | - | - | 5.47 | 37.92 | 7.26 |
| | | Uniform | $\times$ | 49840.8 | 141244.92 | 27822.82 |
| | Magnitude | OWL | $\times$ | 15480.39 | 76684.05 | 21543.82 |
| | | Ours | $\times$ | **8736.22** | **64708.39** | **3487.59** |
| LLaMA2-7B | | Uniform | $\sqrt{}$ | 27.84 | 8557.88 | 30.44 |
| | SparseGPT | OWL | $\sqrt{}$ | 19.71 | 3930.91 | 22.68 |
| | | Ours | $\sqrt{}$ | **18.58** | **677.99** | **19.51** |
| | | Uniform | $\times$ | 76.84 | 778.75 | 78.75 |
| | Wanda | OWL | $\times$ | 30.58 | 450.63 | 36.89 |
| | | Ours | $\times$ | **22.79** | **256.86** | **26.76** |
| | Dense | - | - | 4.88 | 50.94 | 6.72 |
| | | Uniform | $\times$ | 214.19 | 3706.61 | 191.92 |
| | Magnitude | OWL | $\times$ | 57.55 | 2125.47 | 50.79 |
| | | Ours | $\times$ | **52.41** | **1008.90** | **41.89** |
| LLaMA2-13B | | Uniform | $\sqrt{}$ | 19.38 | 450.85 | 23.41 |
| | SparseGPT | OWL | $\sqrt{}$ | 15.12 | 304.91 | 21.74 |
| | | Ours | $\sqrt{}$ | **13.30** | **242.57** | **15.62** |
| | | Uniform | $\times$ | 45.76 | 548.29 | 56.10 |
| | Wanda | OWL | $\times$ | 20.65 | 326.07 | 21.74 |
| | | Ours | $\times$ | **16.19** | **239.34** | **18.47** |

*Table 22.* Accuracy(%) of LLaMA1 on seven zero-shot tasks at 70% unstructured sparsity. The best performance result is indicated in bold.

| Model | Method | Layerwise Sparsity | BoolQ | RTE | HellaSwag | WinoGrande | ARC-e | ARC-c | OBQA | Mean |
|---|---|---|---|---|---|---|---|---|---|---|
| LLaMA1-7B | Dense | - | 75.05 | 66.79 | 76.22 | 70.09 | 72.94 | 44.80 | 44.40 | 64.33 |
| | Magnitude | Uniform | 38.29 | 52.71 | 25.81 | 51.22 | 26.68 | 25.09 | 23.80 | 34.80 |
| | | OWL | 37.83 | 52.71 | 29.34 | **52.80** | 28.24 | 27.65 | 26.20 | 36.40 |
| | | Ours | **38.38** | **53.43** | **35.79** | 52.41 | **31.82** | **28.24** | **27.40** | **38.21** |
| | SparseGPT | Uniform | 65.32 | **54.15** | 42.59 | 58.80 | 41.71 | 26.45 | 28.20 | 45.32 |
| | | OWL | 67.09 | 53.43 | 48.28 | **63.30** | 44.23 | 27.56 | 31.00 | 47.84 |
| | | Ours | **68.62** | 53.43 | **49.13** | 61.33 | **44.61** | **29.35** | **31.80** | **48.32** |
| | Wanda | Uniform | 57.92 | 57.76 | 31.25 | 50.75 | 32.62 | 21.67 | 27.40 | 39.91 |
| | | OWL | 62.57 | 56.68 | 44.28 | 59.59 | 43.69 | 27.05 | 30.40 | 46.32 |
| | | Ours | **63.21** | **59.21** | **47.17** | **60.06** | **50.17** | **28.92** | **31.60** | **48.62** |
| LLaMA1-13B | Dense | - | 77.89 | 70.40 | 79.07 | 72.77 | 74.66 | 47.87 | 44.80 | 66.78 |
| | Magnitude | Uniform | 52.91 | 50.54 | 27.52 | 50.83 | 28.03 | 24.83 | 25.00 | 37.09 |
| | | OWL | 55.87 | 49.10 | **30.27** | 50.43 | **31.48** | 26.62 | 32.40 | 39.45 |
| | | Ours | **58.41** | 50.54 | 29.68 | **51.14** | 30.77 | **27.65** | **32.60** | **40.11** |
| | SparseGPT | Uniform | 67.95 | 52.71 | 48.67 | 61.88 | 46.97 | 28.41 | 31.80 | 48.34 |
| | | OWL | 66.30 | 52.71 | 54.14 | 66.61 | 50.08 | 30.20 | 35.40 | 50.78 |
| | | Ours | **68.47** | **54.87** | **57.92** | **69.14** | **53.70** | **31.74** | **35.60** | **53.06** |
| | Wanda | Uniform | 61.93 | 52.71 | 34.38 | 52.72 | 37.42 | 22.01 | 30.20 | 41.62 |
| | | OWL | 62.78 | 52.71 | 51.08 | 63.06 | 52.31 | 30.20 | 35.00 | 49.59 |
| | | Ours | **66.36** | 52.71 | **56.19** | **64.56** | **56.90** | **31.66** | **35.80** | **52.03** |
| LLaMA1-30B | Dense | - | 82.63 | 66.79 | 82.64 | 75.93 | 78.96 | 52.90 | 48.20 | 69.72 |
| | Magnitude | Uniform | 39.27 | 46.93 | 26.09 | 52.17 | 26.43 | 25.94 | 28.60 | 35.06 |
| | | OWL | 39.02 | **56.68** | 26.30 | 49.33 | 27.86 | 24.49 | 26.40 | 35.73 |
| | | Ours | **61.62** | 47.29 | **34.83** | **52.72** | **39.14** | **27.82** | **30.80** | **42.03** |
| | SparseGPT | Uniform | **68.90** | **57.76** | 60.31 | 69.93 | 61.24 | 35.32 | 37.60 | 55.87 |
| | | OWL | 66.61 | 55.96 | 63.15 | 71.74 | 63.51 | 36.77 | 40.00 | 56.82 |
| | | Ours | 68.87 | 53.79 | **65.68** | **72.38** | **64.48** | **38.65** | **41.00** | **57.84** |
| | Wanda | Uniform | 65.87 | **56.32** | 58.31 | 66.69 | 61.07 | 34.90 | 39.00 | 54.59 |
| | | OWL | **66.30** | 53.79 | 62.12 | 69.69 | 64.06 | 35.32 | 40.20 | 55.93 |
| | | Ours | 64.95 | 48.74 | **65.39** | **70.24** | **67.72** | **38.74** | **42.00** | **56.83** |

*Table 23.* Accuracy(%) of LLaMA2 on seven zero-shot tasks at 70% unstructured sparsity. The best performance result is indicated in bold.

| Model | Method | Layerwise Sparsity | BoolQ | RTE | HellaSwag | WinoGrande | ARC-e | ARC-c | OBQA | Mean |
|---|---|---|---|---|---|---|---|---|---|---|
| LLaMA2-7B | Dense | - | 77.71 | 62.82 | 76.00 | 69.30 | 74.58 | 46.33 | 44.20 | 64.42 |
| | Magnitude | Uniform | 37.95 | **53.07** | 26.36 | 49.33 | 27.86 | 26.96 | 28.00 | 35.65 |
| | | OWL | 40.03 | 52.35 | 30.10 | 48.54 | 30.72 | 26.37 | 27.00 | 36.44 |
| | | Ours | **46.51** | 52.71 | **37.93** | **51.78** | **37.58** | **28.58** | **30.80** | **40.84** |
| | SparseGPT | Uniform | 65.35 | 53.43 | 41.07 | 58.01 | 40.66 | 24.74 | 29.80 | 44.72 |
| | | OWL | 67.92 | 53.07 | 47.97 | 62.04 | 47.31 | 26.02 | 31.80 | 48.02 |
| | | Ours | **71.25** | **53.79** | **50.23** | **62.19** | **49.07** | **27.65** | **33.40** | **49.65** |
| | Wanda | Uniform | 48.23 | 52.71 | 30.28 | 49.96 | 30.30 | 21.42 | 26.40 | 37.04 |
| | | OWL | 62.11 | 52.71 | 37.46 | 56.27 | 42.05 | 24.06 | 30.20 | 43.55 |
| | | Ours | **62.29** | 52.71 | **44.19** | **58.80** | **46.97** | **25.77** | **33.00** | **46.25** |
| LLaMA2-13B | Dense | - | 80.55 | 65.34 | 79.39 | 72.30 | 77.53 | 48.98 | 45.20 | 67.04 |
| | Magnitude | Uniform | 38.62 | 52.71 | 29.56 | 49.41 | 32.11 | 24.57 | 26.60 | 36.23 |
| | | OWL | 38.65 | 52.71 | 43.89 | 54.54 | 37.63 | 28.84 | 28.40 | 40.67 |
| | | Ours | **40.55** | 52.71 | **50.34** | **59.43** | **44.57** | **31.14** | **29.40** | **44.02** |
| | SparseGPT | Uniform | 67.16 | 52.71 | 47.05 | 61.40 | 48.91 | 27.90 | 30.80 | 47.99 |
| | | OWL | 69.45 | **54.87** | 52.86 | 65.27 | 53.24 | 30.38 | 35.80 | 51.70 |
| | | Ours | **74.22** | 54.15 | **55.80** | **65.67** | **54.84** | **33.02** | **36.60** | **53.47** |
| | Wanda | Uniform | 62.11 | 52.71 | 31.71 | 51.78 | 35.73 | 20.82 | 28.20 | 40.44 |
| | | OWL | 63.67 | 52.71 | 46.30 | 60.85 | 51.01 | 28.24 | 34.00 | 48.11 |
| | | Ours | **67.06** | 52.71 | **52.98** | **64.64** | **54.59** | **30.97** | **34.80** | **51.11** |

