# OpenReview forum: "DLP: Dynamic Layerwise Pruning in Large Language Models"
_ICML.cc/2025/Conference — ICML 2025 poster_

### Official Review · Reviewer_zWSE · 2025-02-26

**Overall Recommendation:** 4

**Summary:**

The paper relates to the weight pruning of LLMs and introduces a method to prune individual layers non-homogeneously, i.e. instead of applying a fixed sparsity ratio across all layers, every layer is sparsified according to its importance.

The method extends, and is compatible with previous importance estimation methods, e.g. Magnitude, SparseGPT, WANDA. The method is presented as a competing method to OWL, and is mainly compared against OWL throughout the paper.

According to the authors, the main limitation of OWL is that it relies on a fixed outlier ratio (M), which is problematic as the optimal value of M depends on the model architecture and size, and that even within a given model architecture, they may not be a single optimal value of M for all layers.

The authors propose to do away with the fixed outlier ratio threshold, and instead propose non-parametric relative importance scores for every layer.

The method is backed by extensive empirical results.

## update after rebuttal

I think this is a strong paper that relates to a field that matters to the community, thus I would like to keep my Accept rating.

**Claims And Evidence:**

The claim about OWL's limitation (due to the fixed M value) is justified in Figure 2, which shows that the optimal value of M is highly dependent on the model architecture.

The paper also shows that DLP outperforms OWL according to the LOD metric, despite this being the main optimization criterion of OWL.

An empirical study justifies the particular choice of the function F used in Equation 5, and shows that the median value yields the best results.

**Essential References Not Discussed:**

It would be nice to compare the method with recent works on pruning through distillation (https://arxiv.org/abs/2407.14679) and flexible models (https://arxiv.org/abs/2406.10260).

**Experimental Designs Or Analyses:**

As discussed above, the experimental setup is very sound.

**Methods And Evaluation Criteria:**

The evaluation of the method is quite extensive and applies the method on top of Magnitude, SparseGPT and WANDA pruning, for a variety of models (LLaMA1, LLaMA2, LLaMA3, Vicuna, Mistral, Qwen) and a variety of tasks (language modeling, zero-shot accuracy), and systematically compares results against the uniform baseline, as well as the OWL method.

**Other Comments Or Suggestions:**

line 50: Additionally, we inherently places more emphasis on outliers...  (grammar)

line 200: Howerver, M is ... (typo)

line 265: As shown in Figure 1, OWL and DLP exhibit similar overall trends.  (this is in fact figure 3)

**Other Strengths And Weaknesses:**

The introduction section of the paper presents a detailed overview of the literature, and the style of the paper makes it an enjoyable read.

**Questions For Authors:**

Can you clarify something about equation 5? In my understanding $A_{ij}^l$ is a scalar, yet the $F$ function is meant to operate on a set of values. In practice are you taking the median value of $A_{ij}^l$ over all values of i and j?

**Relation To Broader Scientific Literature:**

The paper provides a lot of interesting insights. The findings from Figure 3 align with other recent works which also show that the first and last layers in an LLM are the hardest to prune.

**Theoretical Claims:**

There is no theoretical claim being made here, as the paper is mainly an empirical study of the proposed method.

---

> ### Author Rebuttal · Authors · 2025-03-31
>
> Dear Reviewer zWSE:
>
>
> We sincerely appreciate your positive assessment of DLP's non-parametric layer importance scoring and the rigor of our experimental design. We are also grateful for your acknowledgment of the paper’s detailed literature review and its clear presentation, which we hope will facilitate broader discussions on layerwise pruning strategies. In the following, we will respond to your comments point by point.
>
>
> >Q1: It would be nice to compare the method with recent works on pruning through distillation (https://arxiv.org/abs/2407.14679) and flexible models (https://arxiv.org/abs/2406.10260).
>
>
> We sincerely appreciate the valuable suggestions from the reviewer. Following your recommendation, we have carefully read the two papers you mentioned.
>
> [[1](https://arxiv.org/abs/2407.14679)] considers multiple dimensions, including depth, width, attention, and MLPs, to perform pruning. It then applies knowledge distillation to retrain on a relatively small dataset, resulting in a powerful and compact model. The paper offers a comprehensive set of best practices for LLM compression and retraining, providing insightful perspectives on practical model pruning.
> FLEXTRON [[2](https://arxiv.org/abs/2406.10260)] dynamically adapts to different latency and accuracy targets during inference without requiring additional fine-tuning. It incorporates input-adaptive routing to maximize performance, significantly reducing training costs.
>
>
> We find both papers highly insightful and valuable. Our paper primarily focuses on non-uniform layerwise pruning strategies, which represent a different contribution compared to these two papers. Since the implementations of these methods are currently not publicly available, we are unfortunately unable to reproduce their experimental results or perform direct comparisons within the limited rebuttal timeframe. We will include citations of these excellent works and discuss their relevance in the revised manuscript. We also look forward to exploring potential integrations with such advanced methods in future research.
>
>
> >Q2: Typos: line 50: Additionally, we inherently places more emphasis on outliers... (grammar);line 200: Howerver, M is ... (typo);line 265: As shown in Figure 1, OWL and DLP exhibit similar overall trends. (this is in fact figure 3)
>
>
> Thank you very much for carefully pointing out these typos. Your suggestions are very helpful for improving the clarity and accuracy of our manuscript. We will correct these errors accordingly in the revised version.
>
>
> >Q3: Can you clarify something about equation 5? In my understanding $\mathbf{A}_ {i j}^l$ is a scalar, yet the F function is meant to operate on a set of values. In practice are you taking the median value of $\mathbf{A}_{i j}^l$ over all values of $i$ and $j$?
>
> Yes, our method is consistent with what you described. For each layer $l$, we aggregate all $\mathbf{A}_ {i j}^l$ values of Linear types into a one-dimensional tensor, compute the median, and then determine the unimportance. Thank you for your question. We will make improvements in the final version to enhance clarity.

---

### Official Review · Reviewer_EVqz · 2025-03-07

**Overall Recommendation:** 2

**Summary:**

The  paper focused on finding important layers to perform non-uniform pruning a.k.a to prune each layer in different rates. More important layers are pruned less aggressively, less important weights pruned more. The paper can be easily extended to any sparsification approach with just determining the pruning rate of each layer.

**Claims And Evidence:**

Author claims:
1. Novel method for measuring layer wise importance that does not rely on empirical values or model type.
2. An effective method for unstructured pruning
3. Wide integration with existing Sparsification methods and PEFT

Second contribution is included in the first one. To the best of my understanding, the paper is not introducing new unstructured pruning method, they are reusing existing ones with varying layer sparsification rate for each layer.

**Essential References Not Discussed:**

The main paper that is missing is EvoPress: https://arxiv.org/pdf/2410.14649. I think the results in there are better than DLP. Technically the paper is from arxiv and it is from October 18, so maybe there is not a requirement to compare with it.

**Experimental Designs Or Analyses:**

Experiments are done thoroughly and there are a lot of empirical analysis of the results. In the paper  latency, throughput and speed up measurement are included in Table 5.

1. Although in appendix: G author provided the set of alpha hyper-parameter, there is no analysis of this parameter: alpha hyper parameter dependence to metrics. This kind of analyze is done for  hyper parameter M in OWL in Figure 2.
2. Table 2: Although the it is important to show results at the Magnitude pruning, PPL higher than 20-30+ is indicator that model were broken in some ways. So comparing ppl 7642.99 to 84511 is meaningless.

**Methods And Evaluation Criteria:**

Yes,  the empirical analysis and the datasets are correctly chosen.

Although, minor comments:
1. I don't get the use of LOD metric. I don't understand what it contribute to?
2. Some of the model are outdated such as llama1, llama2
3. No zero-shot results on llama3 models
4. No results on bigger models f.e. Llama3 70B

**Other Comments Or Suggestions:**

1.Table 2: Although it is important to show results at the Magnitude pruning, PPL higher than 20-30+ is an indicator that model were broken in some ways. So comparing ppl 7642.99 to 84511 is meaningless. Maybe consider moving this this results to appendix and don't discuss it in the main text.

**Other Strengths And Weaknesses:**

Additional strength:
1. Paper is well written and easy to understand, there are a lot of experiments and good discussion about results.
2. Authors provided the code.

Weakness:
1. The paper is very empirical without any theoretical justifications. Basically, this is some criteria that works empirically.
2. Results are not groundbreaking, the improvement although not small at the 70% sparsity level, there are not useful in practice.
For example, improvement from 30 to 22 ppl seems massive, until you start paying attention to the dense model which is 5.4 (See table 3).
At the rate of less than 60% the method is not better than existing ones, please see table 9.

**Questions For Authors:**

1. How your results compare to  EvoPress: https://arxiv.org/pdf/2410.14649?
2. Line 80: DLP reduces the complexity of LLaMA2-7B by 7.79  - what is complexity here?
3. Line 253: Can you please provide formula for LOD and some explanation why this metric matters?
4. What is "m" in Algorithm 1 line 13?
5. How you are choosing alpha and if it is hyper-parameter why it is different of having hyperparameter m in OWL?

**Relation To Broader Scientific Literature:**

The work is based on Wanda criteria. The method can be used on top of other sparsification techniques.

**Theoretical Claims:**

There are no Theoretical Claims in the paper if I didn't miss them.

---

> ### Author Rebuttal · Authors · 2025-03-31
>
> Dear Reviewer EVqz:
>
> We sincerely appreciate your time and thoughtful review of our work. Please find our detailed, point-by-point responses below.
>
> >Q1: Clarify LOD metric and its significance (Line 253) with a formula and explanation.
>
> Thank you for your valuable comment. Followed by OWL, weight outliers are defined as weights with an outlier score at least M times greater than the average. LOD refers to the ratio of weight outliers, calculated as the number of outlier weights divided by the total number of weights (including both zero and non-zero weights), corresponding to Equation 4 in the paper. OWL demonstrates that LOD is positively correlated with model performance. In Table 2 of our paper, we compare the LOD after pruning with OWL. Our method achieves a higher LOD and delivers better performance.
>
> >Q2: Some of the model are outdated such as llama1, llama2; No zero-shot results on llama3 models; No results on bigger models f.e. Llama3 70B.
>
> Thank you for your valuable comment. Due to hardware limitations, we are currently unable to conduct experiments on models larger than 30B parameters. Therefore, we choose the LLaMA1 model family (7B, 13B, and 30B) for our primary experiments. In our response to reviewer AtTG, we present the zero-shot results of LLaMA3-8B using SparseGPT. Additionally, we include two other methods, and the experimental results demonstrate that our approach outperforms the others. We hope to explore evaluations on larger models, such as LLaMA3-70B, in future work once sufficient computational resources become available.
>
> >Q3: Alpha hyper-parameter choice vs. OWL's M
>
> Thank you for your insightful comments. In OWL, weight outliers are defined as weights with an outlier score at least M times greater than the average. In other words, M serves as the threshold for measuring outliers. Our method automatically determines layer importance using the median. Alpha controls the range of layer importance, preventing excessive pruning caused by overly large values, which could lead to significant performance degradation.
>
> >Q4: Limited practical impact despite improvements, underperforming at <60% sparsity compared to existing methods (see Tables 3 and 9).
>
> Thank you for your insightful comments. When hardware resources are limited, such as in edge devices, deploying LLMs with a high number of parameters becomes a challenge. Pruning offers a solution, though it often requires a trade-off in performance to meet hardware constraints. Our method prunes models to 70% sparsity while maintaining reasonable perplexity. Moreover, we demonstrate that the perplexity gap can be significantly narrowed through very short time of fine-tuning. Notably, our approach can also be applied to SVD and PEFT. We believe our method holds promise for integration with future approaches, as long as they allow for layer-wise importance.
>
> >Q5: How your results compare to EvoPress: https://arxiv.org/pdf/2410.14649?
>
> Thank you for your valuable suggestion. Following your advice, we reproduce Evopress and carefully align the relevant parameter settings. Notably, Evopress exhibits higher computational complexity and requires significantly longer pruning times. As shown in the table below, we evaluate the accuracy of LLaMA3-8B on eight zero-shot tasks at 70% sparsity. Experimental results clearly demonstrate that our method consistently outperforms Evopress. We include a detailed discussion of this baseline and its results in the revised version.
>
>
> |  Method  | BoolQ |  RTE  | HellaSwag | WinoGrande | ARC-e | ARC-c | OBQA |  PiQA |    Mean   |
> |:--------:|:-----:|:-----:|:---------:|:----------:|:-----:|:-----:|:----:|:-----:|:---------:|
> | EvoPress | 62.97 | 52.71 |   45.09   |    58.80    | 40.19 | 28.58 | 27.20 | 62.73 |   47.28   |
> |   **Ours**   | 72.54 | 53.07 |   47.17   |    60.14   | 45.16 | 29.18 | 31.20 | 67.25 | **50.71** |
>
> >Q6: Line 80: DLP reduces the complexity of LLaMA2-7B by 7.79 - what is complexity here?
>
> Thank you for pointing out this issue. It actually refers to perplexity, and we will correct it in the final version.
>
>
> >Q7: What is "m" in Algorithm 1 line 13?
>
> Thank you for your valuable comment. "m" represents the mean importance value. To control the overall sparsity while allowing sparsity of each layer to vary according to its importance, we introduce "m" . Its purpose is to centralize the sparsity adjustment, ensuring that the average change in sparsity across layers is zero, thereby maintaining the overall sparsity rate. We will make improvements in the final version to enhance clarity.

---

> > ### Comment · Reviewer_EVqz · 2025-04-07
> >
> > I would like to thank the authors for their detailed response.I still have a few questions left. Sorry for the late reply—I hope you still have time to respond.
> >
> > 1. Q3: So, in your case Alpha is not a hyperparameter that is arbitrarily chosen? Im confused.
> > 2. Q4: Although I agree with your argument in general, in your case, 70% sparsity is not (it is far from) on the Pareto optimality curve. So instead of using 70% sparsity one can just use smaller model with less agressive sparsification. Am I missing something?
> > 3. Q5: Thank you for the experiments. Imho, Including them in your paper would strengthen it.
> > However, one thing is bothering me—the number you reported for EvoPress is much lower than the one in their paper on the same model on the same benchmarks. I would’ve brushed this off as a difference in library or benchmark version, but the difference is too big. Can you please comment on this?

---

> > > ### Author Response · Authors · 2025-04-09
> > >
> > > Dear Reviewer EVqz:
> > >
> > > Thank you for your question and the opportunity to clarify.
> > >
> > > >Q3: So, in your case $\alpha$ is not a hyperparameter that is arbitrarily chosen?
> > >
> > > While $\alpha$ is indeed a hyperparameter in our method, it is not arbitrarily chosen. Instead, we select $\alpha$ within a reasonable range based on empirical observations to ensure a balanced trade-off between pruning aggressiveness and model performance.
> > >
> > > In OWL, two hyperparameters, M and $\lambda$, are controlled simultaneously. M regulates the threshold for outliers—even within the same model, the optimal M value may differ across various parameters (see Figure 2). $\lambda$ determines the range of sparsity to prevent excessive sparsity from causing severe performance degradation.
> > >
> > > In our paper, $\alpha$ serves a role similar to $\lambda$. By constraining the dynamic range of layer importance scores to [0, 2$\alpha$], $\alpha$ ensures that the sparsity allocation preserves inter-layer differences while avoiding extreme pruning. Specifically, we conduct a grid search over $\alpha \in[0.02,0.04, 0.06, 0.08,0.1, 0.12, 0.15, 0.2]$ under different sparsity levels (from 10% to 90%) and select the $\alpha$ value that resulted in the best validation perplexity (see Table 19 in Appendix G).
> > >
> > > >Q4: Although I agree with your argument in general, in your case, 70% sparsity is not (it is far from) on the Pareto optimality curve. So instead of using 70% sparsity one can just use smaller model with less agressive sparsification.
> > >
> > > Thank you for your insightful comment. ​Even when absolute performance is comparable, we believe that pruned models offer distinct advantages.
> > >
> > > **Computational Efficiency:** Pruning reduces model parameters, leading to faster inference and lower resource consumption. Pruned models can utilize specialized sparse inference libraries, enhancing efficiency beyond that of comparably sized models trained from scratch. ​
> > >
> > > **Knowledge Density:** Large models, during pretraining, acquire extensive general knowledge. Pruning retains this knowledge in a compressed form, enabling pruned models to generalize better in data-scarce scenarios compared to smaller models trained from scratch. ​
> > >
> > > **Training Cost:** Pruning leverages existing pretrained models, reducing the need for extensive data collection and prolonged training required for new models. This approach saves computational resources and accelerates deployment. ​
> > >
> > > **Behavioral Consistency:** Pruned models maintain the behavioral patterns of their larger counterparts, ensuring consistent responses and reducing the need for additional safety assessments. In contrast, smaller models trained from scratch may exhibit response deviations.
> > >
> > > In our work, we want to clarify that the demonstration at 70% sparsity serves three purposes: (1) it validates our method’s ability to preserve the critical outlier features essential for LLM functionality even under extreme compression; (2) it enables hardware-friendly deployment through sparsity-aware acceleration—achieving a 3.7× speedup on CPU while retaining compatibility with the original model interface, with Table 7 showing that a model at 70% sparsity can recover reasonable performance after brief fine-tuning. If we fine-tune for a longer period, we believe the performance improves; (3) it reveals new insights into the redundancy patterns of LLMs, insights that may influence future architecture design. In addition, our method extends beyond pruning to include applications in SVD, quantization, and PEFT.
> > >
> > > Therefore, although pruning 70% of a model may not yield optimal performance according to theoretical curves, its advantages in computational acceleration, knowledge retention, cost control and behavioral consistency make this approach highly practical in certain real-world scenarios.
> > >
> > > >Q5: the number you reported for EvoPress is much lower than the one in their paper on the same model on the same benchmarks. I would’ve brushed this off as a difference in library or benchmark version, but the difference is too big.
> > >
> > > Thank you for your insightful comment. In the original EvoPress paper, Fineweb‐Edu serves as a clean and diverse source of calibration data (see Section 4. Experimental Setup). In our reproduction, to ensure fair comparisons and save time, we follow the OWL configuration（see Section 4.1. Main Experiments） and use the same calibration dataset, C4, across all methods. After pruning to 70% sparsity, we evaluate accuracy using the same zero‐shot dataset, with results as detailed in our previous response. We believe that the significant differences observed on the same model and benchmarks may stem from the choice of calibration data, and we maintain that our comparisons remain fair.
> > >
> > > We hope that our explanation above addresses your concerns. We extend our deepest gratitude for your time and expertise in reviewing our work. Your feedback has been invaluable in refining this manuscript.
> > >
> > > Best regards,
> > >
> > > All authors of Submission 1605

---

### Official Review · Reviewer_9Taq · 2025-03-11

**Overall Recommendation:** 4

**Summary:**

This paper aims to improve LLM pruning by introducing a new layerwise sparsity ratio inspired by OWL. Unlike OWL, which assigns layerwise sparsity based on predefined threshold of outlier ratio, DLP measures layerwise importance based on the median of outlier ratios.  This simple modification not only avoids hyperparameter tuning of OWL, but also leads to stronger results. The effectiveness of DLP is extensively evaluated with structured/unstructured llm pruning, low-rank decomposition, quantization, and PEFT following OwLore.

**Claims And Evidence:**

The claims made in the submission are supported by clear and convincing evidence. This paper evaluates their algorithms on almost all perspectives I could think about, just like OWL, including structured/unstructured llm pruning, low-rank decomposition, quantization, and PEFT following OwLore.

**Essential References Not Discussed:**

[1] Li, P., Yin, L. and Liu, S., 2024. Mix-ln: Unleashing the power of deeper layers by combining pre-ln and post-ln. arXiv preprint arXiv:2412.13795.

[2] Sun, Wenfang, Xinyuan Song, Pengxiang Li, Lu Yin, Yefeng Zheng, and Shiwei Liu. "The Curse of Depth in Large Language Models." arXiv preprint arXiv:2502.05795 (2025).

[3] Gromov, Andrey, Kushal Tirumala, Hassan Shapourian, Paolo Glorioso, and Daniel A. Roberts. "The unreasonable ineffectiveness of the deeper layers." arXiv preprint arXiv:2403.17887 (2024).

[4] Men, Xin, Mingyu Xu, Qingyu Zhang, Bingning Wang, Hongyu Lin, Yaojie Lu, Xianpei Han, and Weipeng Chen. "Shortgpt: Layers in large language models are more redundant than you expect." arXiv preprint arXiv:2403.03853 (2024).

**Experimental Designs Or Analyses:**

I am satisfied with the experimental designs.

**Methods And Evaluation Criteria:**

Both perplexity and zero-shot results on commonsense tasks are provided. While the reason behind the success of DLP is not clearly explained, their empirical performance is convincing enough.

I would encourage authors to delve deep to provide intuitions why the median is better than OWL. Why does a low median mean more important layers? The linear decayed importance across layers also strikes me, which is different from my expression. This finding actually aligns well with the recent findings in the ''Curse of Depth'' https://arxiv.org/abs/2502.05795. A plausible understanding is that as the depth increases, deeper layers gradually suffer from large output variance, leading to less contribution to the training. Could the authors provide their intuitions here?

**Other Comments Or Suggestions:**

The paper presentation can be significantly improved.

For instance, Figure 1 can be improved with aligned sub-figures.

Line 20 has no space between text and citation.

Some tables in the appendix lack a bottom line.

**Other Strengths And Weaknesses:**

**Strengths**:

Using median to avoid hyperparameter tuning is important for OWL-like algorithms.

I like the preliminary study in Figure 2, which provides good motivations for the paper.

Evaluating on a wide range of tasks demonstrates the effectiveness of DLP.

**Weaknesses**:

The paper presentation can be significantly improved.

**Questions For Authors:**

I would encourage authors to delve deep to provide intuitions why the median is better than OWL. Why does a low median mean more important layers? The linear decayed importance across layers also strikes me, which is different from my expression. This finding actually aligns well with the recent findings in the ''Curse of Depth'' https://arxiv.org/abs/2502.05795. A plausible understanding is that as the depth increases, deeper layers gradually suffer from large output variance, leading to less contribution to the training. Could the authors provide their intuitions here?

**Relation To Broader Scientific Literature:**

The linear decayed importance across layers also strikes me, which is different from my expression. This finding actually aligns well with the recent findings in recent works [1] [2]. A plausible understanding is that as the depth increases, deeper layers gradually suffer from large output variance, leading to less contribution to the training. Could the authors provide their intuitions here?

[1] Li, P., Yin, L. and Liu, S., 2024. Mix-ln: Unleashing the power of deeper layers by combining pre-ln and post-ln. arXiv preprint arXiv:2412.13795.

[2] Sun, Wenfang, Xinyuan Song, Pengxiang Li, Lu Yin, Yefeng Zheng, and Shiwei Liu. "The Curse of Depth in Large Language Models." arXiv preprint arXiv:2502.05795 (2025).

**Theoretical Claims:**

N/A

---

> ### Author Rebuttal · Authors · 2025-03-31
>
> Dear Reviewer 9Taq:
>
> We sincerely appreciate your recognition of DLP’s median-based importance scoring and its extensive validation across multiple compression paradigms (pruning, quantization, etc.). Below, we address your comments point by point:
> > Q1: The paper presentation can be significantly improved.
>
>
> Thank you for taking the time to review our paper and for your valuable feedback. We will carefully revise and enhance the presentation in the updated manuscript.
>
>
> >Q2: I would encourage authors to delve deep to provide intuitions why the median is better than OWL. Why does a low median mean more important layers? The linear decayed importance across layers also strikes me, which is different from my expression. This finding actually aligns well with the recent findings in the ''Curse of Depth'' https://arxiv.org/abs/2502.05795. A plausible understanding is that as the depth increases, deeper layers gradually suffer from large output variance, leading to less contribution to the training. Could the authors provide their intuitions here?
>
>
> We appreciate your thoughtful comments and particularly thank you for highlighting the connection with recent findings such as the "Curse of Depth". We fully agree with your perspective. Recent research has highlighted the diminishing contribution of deeper layers to learning and representation. For instance, [1] finds that, compared to earlier layers, deeper layers contribute significantly less due to the exponential growth of output variance with model depth, leading to their degradation during training. [2] demonstrates that deeper layers contribute minimally to performance during fine-tuning. Similarly, [3,4] indicate that certain layers have a negligible impact on the overall network function and can be removed without significantly affecting model performance. [5] argues that, ideally, all layers in a model should be sufficiently trained, and feature representations across layers should exhibit enough diversity to maximize the utility of network parameters.
>
>
> Our experiments reveal a similar phenomenon, where deeper layers fail to contribute as effectively as expected. We posit that the median serves as an indicator of layer redundancy. Since the median is insensitive to extreme values, it better captures the central tendency. Elements near the center are easily represented by their neighboring elements, making their removal less detrimental to performance. A lower median within a layer suggests minimal redundancy in its weights, whereas a higher median implies greater redundancy. Consequently, layers with higher redundancy are considered less influential to the overall model and are assigned a higher sparsity rate during pruning.
>
>
> Moreover, the overall increasing trend in sparsity rates suggests that earlier layers are assigned greater importance, likely due to their foundational role in capturing low-level and generalizable features. In contrast, deeper layers often focus on more specialized or task-specific information, which can be more redundant or tolerant to pruning [6]. This aligns with findings in neural network pruning literature, where early layers tend to be more sensitive to pruning than later ones [7,8]. Therefore, our method adaptively allocates pruning budgets based on such structural and functional differences across layers.
>
>
> Thank you for raising such a meaningful question. It inspires us to further explore the fundamental challenges of deeper layers in LLMs in our future work.
>
>
>
>
> [1] Sun W, Song X, Li P, et al. The Curse of Depth in Large Language Models. arXiv preprint arXiv:2502.05795, 2025.
>
> [2] Li P, Yin L, Gao X, et al. Owlore: Outlier-weighed layerwise sampled low-rank projection for memory-efficient llm fine-tuning. arXiv preprint arXiv:2405.18380, 2024.
>
> [3] Gromov A, Tirumala K, Shapourian H, et al. The unreasonable ineffectiveness of the deeper layers. arXiv preprint arXiv:2403.17887, 2024.
>
> [4] Men X, Xu M, Zhang Q, et al. Shortgpt: Layers in large language models are more redundant than you expect. arXiv preprint arXiv:2403.03853, 2024.
>
> [5] Li P, Yin L, Liu S. Mix-ln: Unleashing the power of deeper layers by combining pre-ln and post-ln. arXiv preprint arXiv:2412.13795, 2024.
>
> [6] Fan S, Jiang X, Li X, et al. Not all layers of llms are necessary during inference[J]. arXiv preprint arXiv:2403.02181, 2024.
>
> [7] Chen X, Hu Y, Zhang J, et al. Streamlining redundant layers to compress large language models[J]. arXiv preprint arXiv:2403.19135, 2024.
>
> [8] Yang Y, Cao Z, Zhao H. Laco: Large language model pruning via layer collapse[J]. arXiv preprint arXiv:2402.11187, 2024.

---

### Official Review · Reviewer_AtTG · 2025-03-14

**Overall Recommendation:** 2

**Summary:**

The paper proposes a new method for pruning LLMs by determining the importance of each layer for determining the layer-wise sparsity ratio. The proposed method, DLP, adaptively determines the importance of each layer by combining model weights with input activation information. DLP uses a median to detect outlier values instead of setting a manual threshold value.

**Claims And Evidence:**

The paper claims that median values are used to detect outlier values in the LLM layers. The authors present some evidence by comparing the proposed method with Wanda and SparseGPT, which suggests that the proposed method can achieve better performance at 70\% sparsity; more experiments and comparison against more pruning methods would be required to fully substantiate the claim.

**Essential References Not Discussed:**

Related works are discussed and cited.

**Experimental Designs Or Analyses:**

The method is only compared with SparseGPT and Wanda. More extensive experiments should be conducted and the method should be benchmarked against more recent LLM pruning methods. However, the paper only uses LLaMA family of models, authors should use other families of open-source models to demonstrate the efficacy of the method.

**Methods And Evaluation Criteria:**

Yes,

**Other Comments Or Suggestions:**

There is a typo at line 263.

## update after rebuttal:

I would like to keep my original score. While the authors present extensive experiments, using the median value is not novel enough and there is no theory/analysis why this works.

**Other Strengths And Weaknesses:**

### Strenghts:

1. The paper is well-written and easy to follow.

### Weakness:

1. The technical novelty of the proposed method is limited. The method is a direct extension of previous works and uses median to filter out the outlier values.

2. The method is only compared with two model pruning baselines (SparseGPT, Wanda). Proposed method should be compared against with more recent LLM pruning methods.

**Questions For Authors:**

It is not clear how the overhead timing for the proposed method is lower than uniform pruning in the layer. Could you please provide an explanation.

Do you have any insights as to why the red more or less increases consistently from the layer index in Fig. 6?

**Relation To Broader Scientific Literature:**

The key contribution of the paper is to explore the use of median values instead of mean values to filter out the outlier values and to determine the layerwise sparsity ratio.

**Theoretical Claims:**

No theoretical claim is introduced in the paper.

---

> ### Author Rebuttal · Authors · 2025-03-31
>
> Dear Reviewer AtTG:
>
> Thank you for taking the time to read and review our paper! In the following, we would like to address your comments point by point.
>
> >Q1: Limited technical novelty and direct extension using median-based outlier filtering.
>
> Thank you for your insightful comments. We would like to clarify that our proposed dynamic layerwise importance method is not merely a direct extension. We propose a dynamic layerwise pruning method for LLMs, which automatically evaluates the relative importance of each layer and adaptively assigns non-uniform pruning rates accordingly. The key innovations of our approach are as follows:
>
> (1) **Automatic Layerwise Importance Assessment without Predefined Thresholds:** Unlike OWL, which assigns layerwise sparsity based on a predefined threshold of outlier ratio, our method automatically evaluates layer importance using the median, eliminating the reliance on empirical values and enhancing generalization across different models and parameter settings.
>
> (2)  **Dynamic Adaptability:** By adjusting the pruning rate of each layer based on global relative importance, our approach preserves more parameters in critical layers, significantly improving performance at high sparsity levels (e.g., reducing perplexity by 7.79 and increasing accuracy by 2.7% for LLaMA2-7B at 70% sparsity).
>
> (3) **Compatibility and Scalability:** Our method seamlessly integrates with unstructured pruning, structured pruning, N:M sparsity, quantization, SVD, and PEFT, achieving up to a 3.7x speedup in CPU inference, promoting the practical deployment of LLMs in resource-constrained environments.
>
> Additionally, we clarify that our method emphasizes retaining rather than filtering outliers, as OWL demonstrates that a higher proportion of outliers (LOD) positively correlates with performance. Leveraging the robustness of the median, our approach removes redundancy while effectively preserving crucial outliers, resulting in a higher LOD after pruning (see Table 2).
>
> >Q2: Limited model diversity, only LLaMA.
>
> Thank you for raising this concern. As also noted by Reviewer zWSE, our experiments already include Vicuna, Mistral, and Qwen in addition to the LLaMA series. These models are introduced in Section 4.1 (Experimental Setup), and their results are reported in Section 4.3 (Performance on More Advanced LLMs) and Table 8. We will make this more explicit in the revised manuscript to avoid any confusion.
>
> >Q3: Comparison only with SparseGPT and Wanda, missing other LLM pruning baselines.
>
> Thank you for your valuable comment. In our paper, we compare a range of pruning methods, including Magnitude, Wanda, SparseGPT, OWL, and LLM-Pruner. Based on your suggestion, we have further extended our experiments to include two recent pruning approaches: RIA [1] and AlphaPruning [2]. As shown in the **table 24** (available in the supplementary repository: https://anonymous.4open.science/r/DLP/rebuttal.md ), we evaluate the accuracy of LLaMA3-8B on seven zero-shot tasks at 70% sparsity. Experimental results demonstrate that our method outperforms others using RIA. Moreover, in our response to Reviewer EVqz's Q5, we also provide a comparison with Evopress. We will include such experimental results and discussion in our revised version.
>
>
> [1] Zhang Y, Bai H, Lin H, et al. Plug-and-play: An efficient post-training pruning method for large language models.ICLR 2024.
>
> [2] Lu H, Zhou Y, Liu S, et al. Alphapruning: Using heavy-tailed self regularization theory for improved layer-wise pruning of large language models. NeurIPS 2024.
>
> > Q4: a typo at line 263.
>
> Thank you for pointing this out. We will carefully address this in our revised manuscript.
>
> >Q5: Clarify why the proposed method has lower overhead than uniform pruning.
>
> Thank you for your insightful comment. Our intention in conducting the Pruning Efficiency evaluation is to demonstrate that, although our method incurs slightly more overhead than uniform layerwise pruning due to dynamic layerwise pruning, it does not lead to significant additional empirical pruning time. As shown in Table 6, our approach exhibits comparable time overhead to uniform pruning. In some cases, certain models even run faster with our method, likely because it better aligns with the model’s distribution, allowing for a more efficient selection of pruning units.
>
> >Q6: Do you have any insights as to why the red more or less increases consistently from the layer index in Fig. 6?
>
> Thank you for your insightful comments. We assume you are referring to Fig.3. The overall trend of increasing sparsity indicates that earlier layers are considered more important, while deeper layers are less sensitive to pruning, suggesting that they do not contribute as effectively as expected. For brevity and to avoid redundancy, we have provided a detailed analysis of this topic in our response to Reviewer qTaq's Q2. We are happy to elaborate further if needed.

---

> > ### Comment · Reviewer_AtTG · 2025-04-02
> >
> > Thank you for the detailed response! I missed the experiments on non-llama models during my reviews; apologies about that! My concern with limited novelty remains --- using median value to find pruning ratios is not novel enough. As the reviewer ** EVqz** also noted  "the paper is very empirical without any theoretical justifications. Basically, this is some criteria that works empirically." I would like to keep my original score!

---

> > > ### Author Response · Authors · 2025-04-03
> > >
> > > Dear Reviewer AtTG:
> > >
> > > We sincerely appreciate your insightful feedback and your recognition of our work's empirical effectiveness. We respectfully address the concern regarding novelty and theoretical grounding below:
> > >
> > > While median-based pruning elements exist in prior work (cited in Section 2.3), our key innovation lies in developing a **dynamic layerwise pruning paradigm** that integrates median-driven intra-layer analysis with inter-layer importance modeling. This approach **eliminates the need for predefined thresholds** (e.g., OWL’s empirical outlier criteria)
> > >
> > > While theoretical analysis remains valuable, our work prioritizes solving pressing LLM deployment challenges through a simple, effective, and efficient solution:
> > >
> > > **Simple:** The method requires no complex matrix computations or iterative optimizations (Algorithm 1). Its median-based design ensures straightforward implementation while outperforming sophisticated methods like Evopress[1].
> > >
> > > **Effective:** Extensive experiments across 10+ models (7B-30B), 10 datasets, and 3 pruning paradigms(unstructured pruning, structured pruning and N:M sparsity) demonstrate consistent superiority over SOTA methods, especially at 70% sparsity.
> > >
> > > ​**Efficient:** Achieves 3.7x CPU speedup (Table 5) while maintaining reasonable perplexity at 70% sparsity – a critical threshold for edge deployment.
> > >
> > > Moreover, our work has broader research value.
> > >
> > > ​**Revealing Layer Importance Patterns:** Empirical Study II (Fig 3) reveals a similar phenomenon to recent studies, where deeper layers exhibit lower-than-expected contributions, as mentioned in our response to Reviewer 9Taq's Q2.
> > >
> > >
> > > ​**Enabling Cross-Technique Synergy:** Seamless integration with quantization, SVD, and PEFT (Tables 17-18, Appendix E-F) demonstrates its generalizability beyond basic pruning.
> > >
> > > We agree that theoretical analysis could strengthen the work and will pursue this in future research. However, we believe the method's simplicity, effectiveness, and empirical robustness across diverse scenarios address urgent industry needs for deployable LLM compression – a contribution we humbly submit as valuable to the community.
> > >
> > > Best regards,
> > >
> > > All authors of Submission 1605
> > >
> > > [1] Sieberling O, Kuznedelev D, Kurtic E, et al. Evopress: Towards optimal dynamic model compression via evolutionary search[J]. arXiv preprint arXiv:2410.14649, 2024.

---

### Decision · Program_Chairs · 2025-05-01

**Decision:**

Accept (poster)

**Comment:**

This paper aims to improve LLM pruning via a new layer-wise sparsity ratio inspired by OWL. Unlike OWL, which assigns layer-wise sparsity based on a predefined threshold of outlier ratio, DLP measures layer-wise importance based on the median of outlier ratios. This simple modification not only avoids hyperparameter tuning of OWL, but also leads to stronger results. The effectiveness of DLP is extensively evaluated with structured/unstructured llm pruning, low-rank decomposition, quantization, and PEFT following OwLore.

The paper received mixed reviews, with two reviewers proposing acceptance (score 4), and two reviewers proposing weak rejection (score 2).

Some of the strengths of the paper include:
- The paper is well-written and relatively easy to follow (yet there is room for improvement, and some suggestions for improvements were made).
- Using median to avoid hyper-parameter tuning is important for OWL-like algorithms.
- The preliminary study in Figure 2, used to provide a motivation for the paper, was appreciated.
- Evaluating on a wide range of tasks demonstrates the effectiveness of DLP. Numerous experiments; good discussion.

Some of the weaknesses include:
- The method is a direct extension of previous works and uses median to filter out the outlier values. Hence, the technical novelty is marginal.
- The method is only compared with two model pruning baselines (SparseGPT, Wanda).
- No theoretical insights or results.
- Improvements do not seem to be useful in industrial practice; the work seems mostly conceptual.
- Some missing discussions (ideas for improvements were provided by reviewers); some missing references (again, provided by reviewers).

I have read the reviews, rebuttals and the ensuing discussion. The authors have done a good job responding to the various questions.

In summary, I recommend acceptance.